# INFORMATION FLOW REVEALS WHEN TO TRUST LANGUAGE MODELS

## ABSTRACT

Large language models (LLMs) have emerged as powerful tools for real-world applications, but their utility is often undermined by a fundamental flaw: a tendency toward overconfidence and guessing that leads to unreliable responses. This issue is particularly critical in retrieval-augmented generation (RAG), which is explicitly designed to provide factually grounded answers with retrieved context. Current approaches to quantifying LLM uncertainty are often inadequate, as they rely on surface signals from either the input embeddings or the output space, such as token probabilities or semantic consistency across multiple generations. This work unpacks transformers and assesses response reliability by analyzing the information flow within language models. Specifically, we uncover the contributions of context tokens to the generated output, providing an interpretable basis for evaluating reliability. From this analysis, we introduce two measures. The first, simulatability, assesses the alignment between the context token contributions and their relevance, and the second, concentration, quantifies the extent to which a response's support stems from a narrow subset of tokens. Our experiments demonstrate that these information-flow signals offer a more effective and interpretable basis for assessing reliability than existing methods, outperforming baselines across multiple metrics and advancing the development of more trustworthy LLM deployments. Meanwhile, we also discuss computational considerations and our method's application scope.

## 1 INTRODUCTION

Large language models (LLMs), which are predominantly based on the Transformer architecture (Vaswani et al., 2017), have emerged as a transformative technology for a wide range of applications, including question answering (QA) (Tan et al., 2023), summarization (Zhang et al., 2024), and classification (Howard & Ruder, 2018; Sun et al., 2023). To equip LLMs with up-to-date and domain-specific knowledge, retrieval-augmented generation (RAG) has become a widely adopted paradigm, where relevant documents are retrieved based on the input query and incorporated as additional context to guide generation (Lewis et al., 2020; Guu et al., 2020; Izacard et al., 2023; Shuster et al., 2021). Nevertheless, LLMs can not consistently generate reliable responses, since models often face challenges in accurately identifying the information necessary to answer a given query (Liu et al., 2023). Consequently, effective uncertainty quantification (UQ) is critical in RAG settings, as it enables users to recognize when model outputs are unreliable and to mitigate the risk of incorrect responses.

Most existing UQ methods focus on the output space of LLMs, leveraging logits (Ma et al., 2025), predictive probabilities (Fadeeva et al., 2024), entropy (Malinin & Gales, 2021), or semantic similarity (Kuhn et al., 2023) to assess the reliability of generated outputs (Liu et al., 2025; Shorinwa et al., 2025). However, solely focusing on the output space is insufficient in RAG, where the retrieved context plays a critical role in determining response quality. To overcome this limitation, recent studies have shifted attention to the input space, seeking to quantify the usefulness of retrieved context for a given query (Zhang et al., 2021; Perez-Beltrachini & Lapata, 2025). Nevertheless, both perspectives neglect the internal mechanisms through which LLMs integrate and process retrieved context. This gap motivates a fundamental research question:

*Can the attention mechanism of LLMs be used to assess response uncertainty?*

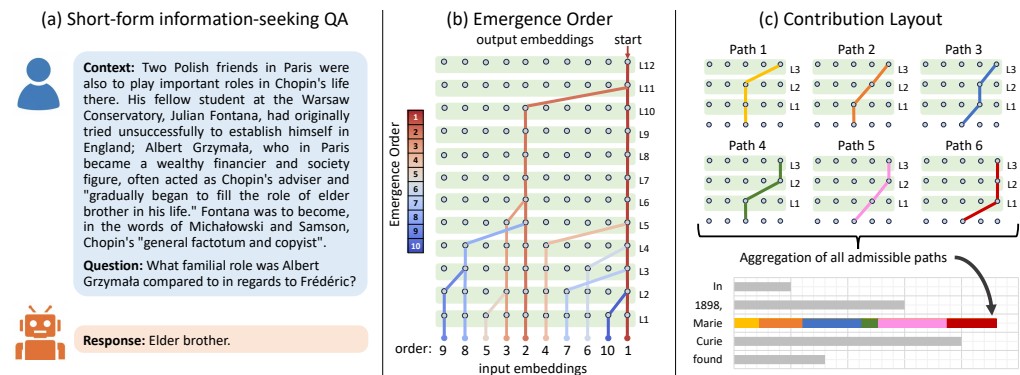

Figure 1: (a) An example of a short-form, information-seeking QA in a RAG system. (b) Principal information flow is extracted in reverse from the model's complete information flow, as detailed in Algorithm 1. The resulting Emergence Order records the sequence of input tokens added to this principal flow, with earlier tokens indicating greater importance for the final generation. For clarity, we neglect MLP operations as they operate independently on each token. (c) Contribution Layout represents the contributions across all input tokens, with each token's contribution defined as the sum of all valid paths from itself to the last token's final embedding.

To address this, we leverage the **information flow** (Ferrando & Voita, 2024) throughout the LLM to quantify the importance of context tokens to the generated response, and propose two quantities:

1. **Emergence Order** captures the sequence in which input tokens are added into the principal information flow, which is extracted from the complete information flow in Algorithm 1. This order highlights the relative importance of input tokens in response generation (Figure 1 (b)).

2. **Contribution Layout** characterizes each input token's contribution, defined as the aggregation of all admissible paths from the token to the final-layer embedding of the last token. This layout offers a holistic view of how input information propagates through the model (Figure 1 (c)).

We further introduce **simulatability**, which compares the two quantities against an estimated relevance layout over the context; higher alignment indicates greater response reliability. Additionally, since the contribution layout can be interpreted as a probability distribution, we quantify **concentration** by comparing it to a uniform distribution over the context tokens using KL divergence. This captures the extent to which contributions are focused on a small subset of tokens, with higher concentration reflecting increased model confidence. Finally, we use the results of these comparisons to optimize a calibrator, enabling more accurate estimation of responses reliability.

We evaluate the proposed method on short-form information-seeking question answering (QA) tasks (Figure 1 (a)) (Rodriguez & Boyd-Graber, 2021). Experiments on SQuAD 2.0 (Rajpurkar et al., 2018) using LLaMA-3.2-3B-Instruct (Grattafiori et al., 2024) demonstrate that our method outperforms existing baselines, achieving an AUROC of 0.75, an AUPRC of 0.83, and an ECE of 0.04.

## 2 RELATED WORK

Prior research on UQ in LLMs can be broadly categorized into two classes. Single-round generation methods primarily rely on token-level predictive probabilities derived from the softmax function (Margatina et al., 2023; Manakul et al., 2023; Fadeeva et al., 2024), entropy computed over the model's output vocabulary (Duan et al., 2023), or by querying the LLM itself to verify the correctness of its output (Kadavath et al., 2022). In contrast, multi-round generation methods estimate uncertainty by evaluating either the semantic consistency (Kuhn et al., 2023; Lin et al., 2023) or the entropy (Malinin & Gales, 2021) across multiple responses produced from a single input. These approaches depend on the assumption that reliable predictions should remain stable under different sampling conditions. In addition, conformal prediction relies on the model's performance on a held-out calibration set, together with the i.i.d. assumption, to derive distribution-free uncertainty guarantees (Kumar et al., 2023; Quach et al., 2023). Yet, uncertainty estimation at the model's output space is inadequate in RAG settings (Yu et al., 2023; Xu et al., 2024; Wang et al., 2024), so recent studies have explored the input space for UQ in RAG, aiming to evaluate the relevance of retrieved context to a given query. For instance, Zhang et al. (2021) leverage context and question embeddings to estimate prediction uncertainty, while Perez-Beltrachini & Lapata (2025) train a calibrator to link the QA model's outputs with the utility of the context in answering the question.

However, these prior works still treat LLMs as black or gray boxes, limiting the ability to understand how context is processed and restricting more fine-grained UQ. Explainable LLMs (Zhao et al., 2024) provide a promising means to understand how language models produce outputs. In this work, we propose a novel UQ method based on information flow (Ferrando et al., 2022; Ferrando & Voita, 2024), leveraging the model's attention mechanisms to evaluate the reliability of its responses.

## 3 BACKGROUND

The computations of multi-head attention[1] within each layer can be equivalently reformulated as a direct expression of the input representations (Kobayashi et al., 2021). Consider a sequence of token embeddings $\mathbf{X} = [\mathbf{x}_1, \dots, \mathbf{x}_T] = [\mathbf{x}_i]_{i=1}^T \in \mathbb{R}^{d \times T}$, where each embedding is $\mathbf{x}_i = \mathbf{X}_{:,i} \in \mathbb{R}^d$. The language model has $H$ heads, each of dimension $d_H = d/H$. For each head $h$, the corresponding query, key, and value projection matrices are denoted as $\mathbf{W}_Q^h$, $\mathbf{W}_K^h$, and $\mathbf{W}_V^h \in \mathbb{R}^{d_H \times d}$.

The query, key, and value vectors for embedding $\mathbf{x}_i$ are

$$\mathbf{q}_i^h = \mathbf{W}_Q^h \mathbf{x}_i \in \mathbb{R}^{d_H}, \quad \mathbf{k}_i^h = \mathbf{W}_K^h \mathbf{x}_i \in \mathbb{R}^{d_H}, \quad \mathbf{v}_i^h = \mathbf{W}_V^h \mathbf{x}_i \in \mathbb{R}^{d_H}. \tag{1}$$

The attention weight between $\mathbf{x}_i$ and $\mathbf{x}_j$ in head $h$ is defined as

$$\mathbf{A}_{i,j}^h = \frac{\exp(\langle \mathbf{q}_i^h, \mathbf{k}_j^h \rangle / \sqrt{d_H})}{\sum_{t=1}^T \exp(\langle \mathbf{q}_i^h, \mathbf{k}_t^h \rangle / \sqrt{d_H})} \in \mathbb{R}. \tag{2}$$

Accordingly, the output of head $h$ for input $\mathbf{x}_i$ is $\mathbf{z}_i^h = \sum_{j=1}^T \mathbf{A}_{i,j}^h \mathbf{v}_j^h \in \mathbb{R}^{d_H}$. Concatenating the outputs of all heads and projecting through $\mathbf{W}_O \in \mathbb{R}^{d \times d}$ yields $\mathbf{W}_O \cdot \text{Concat}(\mathbf{z}_i^1, \dots, \mathbf{z}_i^H) \in \mathbb{R}^d$. Equivalently, partitioning $\mathbf{W}_O$ into submatrices $\mathbf{W}_O^h \in \mathbb{R}^{d \times d_H}$ allows to express the projection as a summation. Finally, incorporating the residual connection gives

$$\mathbf{y}_i = \mathbf{x}_i + \sum_{h=1}^H \mathbf{W}_O^h \mathbf{z}_i^h \in \mathbb{R}^d. \tag{3}$$

This allows us to define an attribution vector from input $\mathbf{x}_j$ to the output embedding $\mathbf{y}_i$ by

$$a(\mathbf{y}_i, \mathbf{x}_j) = \mathbf{1}_{\{j=i\}} \mathbf{x}_i + \sum_{h=1}^H \mathbf{W}_O^h \mathbf{A}_{i,j}^h \mathbf{W}_V^h \mathbf{x}_j \in \mathbb{R}^d, \tag{4}$$

where $\mathbf{1}_{\{j=i\}}$ is an indicator function that equals 1 if $j = i$ and 0 otherwise. The contribution from $\mathbf{x}_j$ to the output $\mathbf{y}_i$ is measured by the normalized Manhattan similarity (Ferrando et al., 2022)

$$\text{dist}(\mathbf{y}_i, a(\mathbf{y}_i, \mathbf{x}_j)) = \frac{\max(0, \|\mathbf{y}_i\|_1 - \|\mathbf{y}_i - a(\mathbf{y}_i, \mathbf{x}_j)\|_1)}{\sum_{t=1}^T \max(0, \|\mathbf{y}_i\|_1 - \|\mathbf{y}_i - a(\mathbf{y}_i, \mathbf{x}_t)\|_1)} \in \mathbb{R}. \tag{5}$$

Based on Eq. (5), we build a contribution matrix $\mathbf{C} \in \mathbb{R}^{T \times T}$ where the $(i,j)$-th entry $\mathbf{C}_{i,j} = \text{dist}(\mathbf{y}_i, a(\mathbf{y}_i, \mathbf{x}_j))$. Since attention in causal language models is autoregressive, each token can only depend on itself and its predecessors. Hence, $\mathbf{A}_{i,j}^h = 0$ if $j > i$. As a result, $\mathbf{C}$ is lower-triangular with all entries above the main diagonal equal to zero. For a model with $L$ transformer layers, we compute a contribution matrix at each layer, denoted $\mathbf{C}^{(l)}$ for $l = 1, \dots, L$. The input and output embeddings of the $i$-th token at layer $l$ are denoted by $\mathbf{x}_i^{(l)}$ and $\mathbf{y}_i^{(l)}$, respectively. The collection of these layer-wise matrices, $\{\mathbf{C}^{(l)}\}_{l=1}^L$, is the complete information flow through the model. Figure 2 visualizes the matrices, where a consistent color is used for elements and flows targeting the same token.

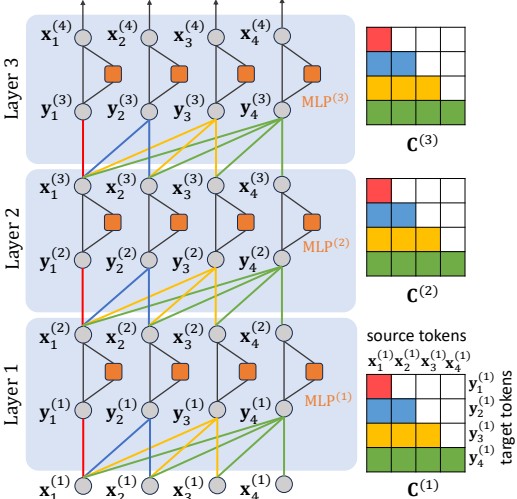

Figure 2: Layer-wise contribution matrices.

---

[1]We omit the MLP and layer normalization operations here because they operate independently on each token and do not affect the inter-token interactions we focus on.

# 4 METHOD

Using the layer-wise contribution matrices $\{\mathbf{C}^{(l)}\}_{l=1}^L$, we define Emergence Order and Contribution Layout to measure the relative importance of input tokens to the model's output.

## 4.1 EMERGENCE ORDER

As the model relies solely on the final-layer representation of the last input token to predict the next token, Ferrando & Voita (2024) start from this representation to extract a subflow from $\{\mathbf{C}^{(l)}\}_{l=1}^L$ in a backward manner, using a pre-specified threshold. Input tokens included in the subflow are considered important for the generation. However, this binary criterion merely separates tokens into 'important' and 'unimportant' classes based on the fixed threshold, providing no continuous measure of relative importance.

To address the limitations, we introduce the **Auto-Emergence** algorithm. This algorithm extracts principal information flows, denoted as $\{\mathbf{P}^{(l)}\}_{l=1}^L$, from the complete layer-wise contribution matrices $\{\mathbf{C}^{(l)}\}_{l=1}^L$. The process produces a vector $\mathbf{E} \in \mathbb{R}^T$ that records the **Emergence Order** of input tokens, where a token's earlier position reflects its greater relative significance.

The algorithm begins at the final layer $L$ with the last input token's output embedding $\mathbf{y}_T^{(L)}$, whose self-contributions $\mathbf{C}_{T,T}^{(l)}$ are extracted into the corresponding principal flow element $\mathbf{P}_{T,T}^{(l)}$ for $l = 1, ..., L$, since self-contributions are typically dominant due to residual connections. We then assign $\mathbf{E}_T = 1$ and create a selection pool $\mathcal{S}$, which consists of all flows connected to the extracted ones.

Subsequent extraction is an iterative top-down search. At each step, we extract the strongest flow from the pool $\mathcal{S}$. When a flow $\mathbf{C}_{i,j}^{(k)}$ is incorporated, its self-contributions from preceding layers, $\mathbf{C}_{j,j}^{(l)}$ for $l < k$, are also extracted. $\mathbf{E}_j$ is then assigned the next available rank and $\mathcal{S}$ will be updated.

The process continues until all tokens are ranked. The vector $\mathbf{E}$ reveals how the input tokens influence the generation process. The workings of the method are illustrated with an example in Figure 3, and its steps are detailed in Algorithm 1.

---

**Algorithm 1** Auto-Emergence Algorithm

**Require:** contribution matrices $\{\mathbf{C}^{(l)}\}_{l=1}^L$

**Initialization:**
$\quad \mathbf{P}^{(l)} \leftarrow 0 \in \mathbb{R}^{T \times T} \quad \forall l = 1, \ldots, L$
$\quad \mathbf{E} \leftarrow 0 \in \mathbb{R}^T$
$\quad \mathbf{E}_T \leftarrow 1$
$\quad \mathbf{P}_{T,T}^{(l)} \leftarrow \mathbf{C}_{T,T}^{(l)} \quad \forall l = 1, \ldots, L$
$\quad \mathcal{S} \leftarrow \{\mathbf{C}_{T,j}^{(l)} \mid j < T, l \leq L\}$

**Extraction:**
**while** $\exists i$ s.t. $\mathbf{E}_i = 0$ **do**
$\quad$ Select $\mathbf{C}_{i,j}^{(k)} = \mathrm{argmax}(\mathcal{S})$
$\quad \mathbf{P}_{i,j}^{(k)} \leftarrow \mathbf{C}_{i,j}^{(k)}$
$\quad \mathbf{P}_{j,j}^{(l)} \leftarrow \mathbf{C}_{j,j}^{(l)}$ for $l = 1, \ldots, k-1$
$\quad \mathbf{E}_j \leftarrow \min\{r \in \mathbb{Z}^+ \mid r \notin \mathbf{E}\}$
$\quad \mathcal{S} \leftarrow \mathcal{S} \cup \{\mathbf{C}_{j,m}^{(l)} \mid m < j, l \leq k-1\}$
**end while**
**return** $\{\mathbf{P}^{(l)}\}_{l=1}^L, \mathbf{E}$

---

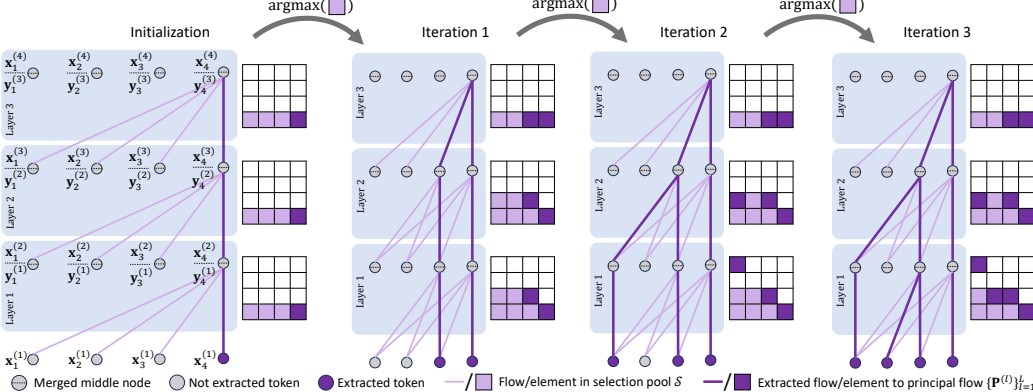

Figure 3: Demonstration of the Auto-Emergence algorithm. We extract the principal information flow from the layer-wise contribution matrices $\{\mathbf{C}^{(l)}\}_{l=1}^L$. The Emergence Order of the example is $\mathbf{E} = [3, 4, 2, 1]$. MLPs operating independently on each token are merged within the middle nodes to simplify the representation.

### 4.2 CONTRIBUTION LAYOUT

The collection of complete layer-wise contribution matrices $\{\mathbf{C}^{(l)}\}_{l=1}^{L}$ characterizes local information flow within each transformer layer. However, these matrices are high-dimensional and difficult to interpret directly. To obtain a compact description of how input tokens influence the final-layer representations, we compose layer-wise contributions into a single total contribution matrix:

$$\mathbf{C}^{\text{total}} = \mathbf{C}^{(L)}\mathbf{C}^{(L-1)}\cdots\mathbf{C}^{(1)} \in \mathbb{R}^{T \times T}. \tag{6}$$

The entry $\mathbf{C}_{i,j}^{\text{total}}$ measures the overall contribution from input embedding $\mathbf{x}_j^{(1)}$ to the output representation $\mathbf{y}_i^{(L)}$. It can be expressed as a weighted sum over all valid paths that connect the $j$-th token at the input layer to the $i$-th token at the final layer:

$$\mathbf{C}_{i,j}^{\text{total}} = \sum_{s_{L-1}=j}^{i} \sum_{s_{L-2}=j}^{s_{L-1}} \cdots \sum_{s_1=j}^{s_2} \mathbf{C}_{i,s_{L-1}}^{(L)} \mathbf{C}_{s_{L-1},s_{L-2}}^{(L-1)} \cdots \mathbf{C}_{s_1,j}^{(1)} \in \mathbb{R}. \tag{7}$$

The summation indices $(s_1,\ldots,s_{L-1})$ enumerate all admissible intermediate tokens along the path from source $j$ to target $i$. The indices in Eq. (7) are implicitly subject to the monotonicity constraint

$$j \leq s_1 \leq s_2 \leq \cdots \leq s_{L-1} \leq i, \tag{8}$$

which ensures that information flows consistently "upstream," never skipping or exceeding the valid range between $i$ and $j$. This constraint arises naturally from the causal masking in autoregressive transformers: a target token can only attend to its preceding tokens, and a source token can only influence subsequent tokens. Under this constraint, each valid sequence of indices defines one admissible path through the network from $j$ to $i$, with the path's strength determined by the product of per-layer contributions along it.

For example in Figure 4, for a two-layer model ($L = 2$), the total contribution from input $\mathbf{x}_1^{(1)}$ to the final-layer output $\mathbf{y}_3^{(2)}$ is

$$\mathbf{C}_{3,1}^{\text{total}} = \sum_{s_1=1}^{3} \mathbf{C}_{3,s_1}^{(2)} \mathbf{C}_{s_1,1}^{(1)}$$

$$= \mathbf{C}_{3,1}^{(2)}\mathbf{C}_{1,1}^{(1)} + \mathbf{C}_{3,2}^{(2)}\mathbf{C}_{2,1}^{(1)} + \mathbf{C}_{3,3}^{(2)}\mathbf{C}_{3,1}^{(1)},$$

where each term corresponds to a distinct path via $\mathbf{y}_1^{(1)}$, $\mathbf{y}_2^{(1)}$, and $\mathbf{y}_3^{(1)}$, respectively. Thus, $\mathbf{C}^{\text{total}}$ consolidates the entire information flow into a single interpretable matrix, while preserving the underlying path semantics across layers. Since only the last

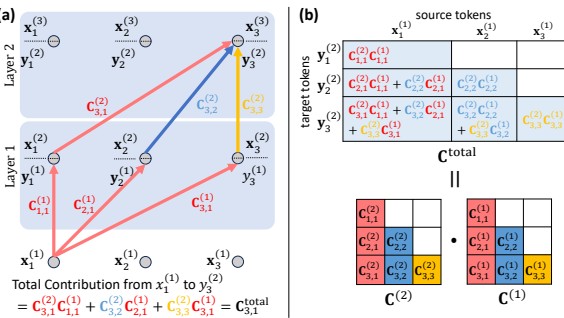

Figure 4: (a) Admissible paths from the first to third token in a two-layer transformer. (b) Computation of the total contribution matrix $\mathbf{C}^{\text{total}}$. Flows and matrix elements from the same source share the same color per layer.

token's output representation is used to generate the next token, the **Contribution Layout** is given by the last row of $\mathbf{C}^{\text{total}}$, capturing the influence of all input tokens on the generated token:

$$\mathbf{C}^{\text{layout}} = \mathbf{C}_{-1,:}^{\text{total}} \in \mathbb{R}^{T}. \tag{9}$$

While $\mathbf{C}^{\text{layout}}$ captures the influence of all input tokens on the generated token, it often contains small, noisy flows from less relevant paths. To address this, we leverage the principal flows $\{\mathbf{P}^{(l)}\}_{l=1}^{L}$ obtained from Algorithm 1 to compute a principal contribution layout:

$$\mathbf{P}^{\text{total}} = \mathbf{P}^{(L)}\mathbf{P}^{(L-1)}\cdots\mathbf{P}^{(1)} \in \mathbb{R}^{T \times T}, \quad \mathbf{P}^{\text{layout}} = \mathbf{P}_{-1,:}^{\text{total}} \in \mathbb{R}^{T}, \tag{10}$$

which provides a more interpretable representation of the dominant contributions to generation.

### 4.3 CONTEXT SLICING ACROSS MULTIPLE GENERATIONS

In RAG, the input sequence typically consists of an instruction prompt, a context, and a question, in that order. Let the total number of input tokens be $T = T_p + T_c + T_q$, where $T_p$, $T_c$, and $T_q$ denote the lengths of the instruction prompt, context, and question, respectively. The emergence order $\mathbf{E}$ and the contribution layouts $\mathbf{C}^{\text{layout}}$, $\mathbf{P}^{\text{layout}}$ are defined over the full sequence of $T$ input tokens.

Since our primary interest lies in how LLMs process the retrieved context, we focus on the context segment of the input sequence. Formally, we define the index list $\mathbf{I} = [T_p + 1, ..., T_p + T_c]$, which corresponds to the token positions allocated to the context. For a single generated token, the sliced emergence order and contribution layouts are then defined as

$$\mathbf{E_I} := [\mathbf{E}_i]_{i \in \mathbf{I}} \in \mathbb{R}^{T_c}, \quad \mathbf{C_I}^{\text{layout}} := [\mathbf{C}_i^{\text{layout}}]_{i \in \mathbf{I}} \in \mathbb{R}^{T_c}, \quad \mathbf{P_I}^{\text{layout}} := [\mathbf{P}_i^{\text{layout}}]_{i \in \mathbf{I}} \in \mathbb{R}^{T_c}. \quad (11)$$

When generating multiple tokens, we obtain a sequence of context emergence order $\{\mathbf{E_I}(t)\}_{t=1}^{T_g}$ and contribution layouts $\{\mathbf{C_I}^{\text{layout}}(t)\}_{t=1}^{T_g}$, $\{\mathbf{P_I}^{\text{layout}}(t)\}_{t=1}^{T_g}$, where $T_g$ is the number of generated tokens. We then aggregate over all generated tokens to form averaged results in $\mathbb{R}^{T_c}$:

$$\bar{\mathbf{E}}_\mathbf{I} = \frac{1}{T_g} \sum_{t=1}^{T_g} \mathbf{E_I}(t), \quad \bar{\mathbf{C}}_\mathbf{I}^{\text{layout}} = \frac{1}{T_g} \sum_{t=1}^{T_g} \mathbf{C_I}^{\text{layout}}(t), \quad \bar{\mathbf{P}}_\mathbf{I}^{\text{layout}} = \frac{1}{T_g} \sum_{t=1}^{T_g} \mathbf{P_I}^{\text{layout}}(t). \quad (12)$$

### 4.4 CALIBRATED CONFIDENCE OF LLM RESPONSES IN RAG

#### 4.4.1 RELEVANCE LAYOUT

To establish a reference for evaluation, we use Qwen-3-Reranker-8B (Zhang et al., 2025), which assigns a single scalar score, denoted by $r$, representing the overall relevance of a context to a given question. While this score reflects the model's holistic judgment, it does not reveal the role of individual context tokens in making the decision. To decompose this relevance score into token-level signals, we make use of Shapley values, a well-established concept from cooperative game theory (Lundberg & Lee, 2017b; Sundararajan & Najmi, 2020). The key intuition is to treat each token in the context as a "player" in a cooperative game, where the total payoff is the relevance score $r$. The Shapley framework then assigns each token a fair share of this payoff by averaging its marginal effect across all possible subsets of tokens. From these token-level attributions, we construct the **Relevance Layout**, denoted by $\mathbf{R}^{\text{layout}} \in \mathbb{R}^{T_c}$, to provide a fine-grained view of how the reranker assesses the usefulness of each token for answering the question. It serves as an estimated ground truth against which we compare the model-derived results in our UQ framework.

#### 4.4.2 FIDELITY OF MODEL-DERIVED LAYOUTS

**Simulatability.** We introduce simulatability as a measure of how well the model's internal context processing aligns with an external notion of token relevance. Intuitively, if the emergence order $\bar{\mathbf{E}}_\mathbf{I}$ and the contribution layouts $\bar{\mathbf{C}}_\mathbf{I}^{\text{layout}}$ and $\bar{\mathbf{P}}_\mathbf{I}^{\text{layout}}$ place emphasis on the same tokens as the estimated relevance layout $\mathbf{R}^{\text{layout}}$, then the model's reasoning process and response are more reliable.

The comparison starts by ranking context tokens. Since $\mathbf{R}^{\text{layout}}$, $\bar{\mathbf{C}}_\mathbf{I}^{\text{layout}}$, and $\bar{\mathbf{P}}_\mathbf{I}^{\text{layout}} \in \mathbb{R}^{T_c}$ assign a real-valued score to each context token, with larger values indicating higher importance, we transform them into index lists by sorting their values in descending order. In contrast, $\bar{\mathbf{E}}_\mathbf{I} \in \mathbb{R}^{T_c}$ encodes an emergence order: smaller values indicate earlier entry into the principal flow in Algorithm 1, and thus greater importance. Accordingly, we sort its indices in ascending order. Formally, denoting $\pi$ a permutation of context token indices, we have

$$\pi_\mathbf{R} = \text{argsort}_\downarrow(\mathbf{R}^{\text{layout}}), \quad \pi_\mathbf{C} = \text{argsort}_\downarrow(\bar{\mathbf{C}}_\mathbf{I}^{\text{layout}}), \quad \pi_\mathbf{P} = \text{argsort}_\downarrow(\bar{\mathbf{P}}_\mathbf{I}^{\text{layout}}), \quad \pi_\mathbf{E} = \text{argsort}_\uparrow(\bar{\mathbf{E}}_\mathbf{I}).$$

We evaluate simulatability by comparing $\pi_\mathbf{C}$, $\pi_\mathbf{P}$, and $\pi_\mathbf{E}$ against $\pi_\mathbf{R}$ using **rank-biased overlap (RBO)** (Webber et al., 2010), which emphasizes agreement at higher-ranked tokens. Specifically, we compute $\text{RBO}(\pi_\mathbf{C}, \pi_\mathbf{R})$, $\text{RBO}(\pi_\mathbf{P}, \pi_\mathbf{R})$, and $\text{RBO}(\pi_\mathbf{E}, \pi_\mathbf{R})$ to quantify the alignment of each ranking with $\pi_\mathbf{R}$. The computation is governed by a persistence parameter $p$, with larger values giving more weight to lower-ranked items and smaller values emphasizing higher-ranked items. In our experiments, we set $p = 0.7$. Detailed computation procedures are provided in Appendix C.

**Concentration.** Understanding how focused a model's reasoning is across context tokens can reveal whether the model relies on a few key context tokens or distributes contributions broadly. To quantify this, we examine the concentration of both $\bar{\mathbf{C}}_\mathbf{I}^{\text{layout}}$ and $\bar{\mathbf{P}}_\mathbf{I}^{\text{layout}}$. Highly concentrated layouts indicate that a small subset of tokens dominates the model's internal computation, reflecting strong confidence, whereas uniform layouts suggest that the model pays attention to tokens evenly.

Since both $\bar{\mathbf{C}}_{\mathbf{I}}^{\text{layout}}$ and $\bar{\mathbf{P}}_{\mathbf{I}}^{\text{layout}}$ assign positive real-valued scores to each context token, denoting $\Delta^{T_c-1}$ the standard simplex in $\mathbb{R}^{T_c}$, we first normalize them onto the probability simplex:

$$\widehat{\mathbf{C}}_{\mathbf{I}}^{\text{layout}} = \frac{\bar{\mathbf{C}}_{\mathbf{I}}^{\text{layout}}}{\sum_{i\in\mathbf{I}} \bar{\mathbf{C}}_i^{\text{layout}}} \in \Delta^{T_c-1}, \quad \widehat{\mathbf{P}}_{\mathbf{I}}^{\text{layout}} = \frac{\bar{\mathbf{P}}_{\mathbf{I}}^{\text{layout}}}{\sum_{i\in\mathbf{I}} \bar{\mathbf{P}}_i^{\text{layout}}} \in \Delta^{T_c-1}.$$

The uniform distribution over $T_c$ tokens is defined as $\mathbf{U} = [\frac{1}{T_c}, \frac{1}{T_c}, \ldots, \frac{1}{T_c}]$. We quantify the deviation of the layouts from uniformity using **Kullback-Leibler (KL) divergence** (Van Erven & Harremos, 2014), denoted by $\text{KL}(\widehat{\mathbf{C}}_{\mathbf{I}}^{\text{layout}} \| \mathbf{U})$ and $\text{KL}(\widehat{\mathbf{P}}_{\mathbf{I}}^{\text{layout}} \| \mathbf{U})$. KL divergence measures the information-theoretic discrepancy between the observed layouts and uniformity. It is particularly sensitive to sharp peaks in the distribution, thereby highlighting concentration on a small subset of tokens. The computation of KL divergence for discrete distributions is provided in Appendix D.

### 4.4.3 CALIBRATOR FOR RESPONSE CONFIDENCE

We develop a multi-level granularity that groups context tokens into word- and phrase-level units for computing emergence order and contribution layouts, which are then used to measure simulatability (see Appendix E for details). In addition, we use the scalar score $r$ from Qwen-3-Reranker-8B as an indicator of overall context relevance: higher values of $r$ indicate that the context is more pertinent to the question and more informative for the model, increasing the reliability of the generated response.

Finally, we combine these features—simulatability, concentration, and the context relevance score $r$—to train a calibrator that outputs a calibrated confidence for the model's response. Together, these features enable the calibrator to provide confidence estimates that reflect both the model's internal reasoning dynamics and the quality of the retrieved context. The discriminative power of each feature is evaluated in Appendix F.

## 5 EXPERIMENT

### 5.1 EXPERIMENTAL SETUP

**Dataset.** We conduct experiments on the SQuAD2.0 dataset (Rajpurkar et al., 2018), from which we randomly sample 42,000 examples. Each example consists of a context, a question, and a corresponding ground-truth answer, if available. For methods that require training a calibrator, the data are split into training, validation, and test sets with a 3:1:1 ratio. For methods that do not require calibration, only the test split is used for evaluation.

**Model Selection.** We use LLaMA-3.2-3B-Instruct (Grattafiori et al., 2024) as the base question answering (QA) model. To ensure the model focuses on short-form information-seeking questions, we instruct the model to generate responses containing at most five words. Specifically, we provide the model with an input sequence as follows:

> Answer the question in no more than five words.
> Context: {context} Question: {question} Answer:

Here, context and question are placeholders that are replaced with the retrieved passage and the corresponding query from the SQuAD 2.0 dataset, respectively. A concrete example of the input format is provided in Appendix I. To determine if a predicted response is correct, we avoid token-level overlap metrics (e.g., BERTScore (Zhang et al., 2020)) and adopt a semantic evaluation pipeline. Specifically, we merge the predicted answer and the ground truth with the corresponding question into two natural language statements using Qwen2.5-7B (Qwen et al., 2025). The resulting statements are then compared by HHEM-2.1-Open (Bao et al., 2024), which assigns a similarity score ranging from 0 to 1. We provide a concrete example illustrating how prediction correctness is determined in Appendix I. A prediction is labeled "incorrect" if the similarity score falls below 0.5. Importantly, SQuAD2.0 contains unanswerable questions whose associated contexts lack the necessary information. In such cases, the QA model is expected to acknowledge the insufficiency of evidence explicitly. To evaluate this behavior, we compare the model's response against a set of predefined candidates (e.g., "I do not know.", "It is not mentioned.") using HHEM-2.1-Open.

The calibrator introduced in Section 4.4.3 is trained with XGBoost library (Chen & Guestrin, 2016). Hyperparameter optimization is performed on the validation set using Optuna (Akiba et al., 2019).

**UQ Baselines.** We select a set of standard UQ frameworks for LLMs that complement each other by covering different aspects of uncertainty estimation. The majority of existing methods operate in the output space, leveraging either a single forward pass or multiple generations. In the single-generation category, Perplexity (PPL) (Margatina et al., 2023) quantifies uncertainty by measuring predicted softmax probabilities of the output tokens, while P(True) (Kadavath et al., 2022) assesses the correctness of a prediction by querying the QA model itself to validate its answer. In contrast, multi-generation methods assess uncertainty by aggregating multiple outputs for the same input: Regular Entropy (Malinin & Gales, 2021) averages the predictive entropy of these outputs, and Semantic Entropy (Kuhn et al., 2023) measures consistency in the semantic content among them. To account for the retrieval context unique to RAG, we further include UQ approaches that incorporate the input space. KnowingMore (Zhang et al., 2021) integrates the embeddings of both context and question to calibrate prediction confidence, while Utility Ranker (Perez-Beltrachini & Lapata, 2025) directly estimates the usefulness of retrieved context for answering the question.

**Evaluation Metrics.** The efficacy of all methods, including the proposed approach, is assessed using a suite of discriminative metrics. Specifically, we employ the AUPRC and AUROC to quantify a method's ability to discriminate between positive and negative instances, with a higher value indicating superior performance in ranking positive cases above negative ones. Since P(True), KnowingMore, and our method offer calibrated confidence, we provide additional evaluations of their calibration performance. These include the calibration accuracy at a confidence threshold of 0.5 and expected calibration error (ECE). These metrics are crucial for applications where a certain level of confidence or reliability is a prerequisite for deployment. Finally, we compute Spearman and Pearson correlation coefficients to quantify the relationship between the similarity scores from HHEM-2.1-Open and the UQ scores from each method. A higher positive correlation indicates that the UQ scores are more consistent with the judgment of HHEM-2.1-Open, reflecting better alignment between the model's uncertainty estimates and the actual quality of the predictions. This avoids the bias that can arise from using a fixed threshold to classify predictions as correct or incorrect.

## 5.2 RESULTS

As shown in Table 1, our proposed method achieves the highest discriminative capability, with an AUROC of 0.75 and an AUPRC of 0.83, surpassing all baseline approaches. For all methods, the AUPRC values consistently exceed the corresponding AUROC values. This occurs because AUPRC emphasizes performance on the positive class and is more sensitive to class imbalance, whereas AUROC considers both positive and negative classes equally. Since the LLaMA-3.2-3B-Instruct model performs well on the SQuAD 2.0 dataset, we have a relatively larger number of correctly predicted instances, which leads to higher AUPRC values compared with AUROC.

Regarding calibration, our method also demonstrates the highest reliability among all evaluated approaches, achieving a calibration accuracy of 0.73. In addition, it attains the lowest expected calibration error (ECE) of 0.04, which is computed by partitioning the $[0, 1]$ confidence interval into 10 equal-width sub-intervals. These results indicate that the predicted confidence scores from our method are closely aligned with the empirical probability of correctness, reflecting not only accurate discrimination but also reliable confidence estimation. For correlation with HHEM-2.1-Open, our approach shows the strongest alignment, with a Spearman coefficient of 0.39 and a Pearson coefficient of 0.45. This suggests that our uncertainty estimates capture patterns of correctness that correspond closely to the assessments made by HHEM-2.1-Open. Additional experiment results with different inference models and datasets are shown in Appendix B.

Table 1: Uncertainty quantification performance of different methods on the SQuAD 2.0 dataset evaluated with the LLaMA-3.2-3B-Instruct model. Bold indicates the best value for each metric.

| Method | AUROC ↑ | AUPRC ↑ | Cali. Acc ↑ | ECE ↓ | Spearman ↑ | Pearson ↑ |
|---|---|---|---|---|---|---|
| PPL | 0.62 | 0.77 | / | / | 0.19 | 0.21 |
| P(True) | 0.57 | 0.71 | 0.53 | 0.16 | 0.13 | 0.08 |
| Regular Entropy | 0.72 | 0.81 | / | / | 0.36 | 0.39 |
| Semantic Entropy | 0.71 | 0.78 | / | / | 0.32 | 0.36 |
| KnowingMore | 0.69 | 0.81 | 0.65 | 0.22 | 0.30 | 0.26 |
| Utility Ranker | 0.66 | 0.77 | / | / | 0.28 | 0.22 |
| Ours | **0.75** | **0.83** | **0.73** | **0.04** | **0.39** | **0.45** |

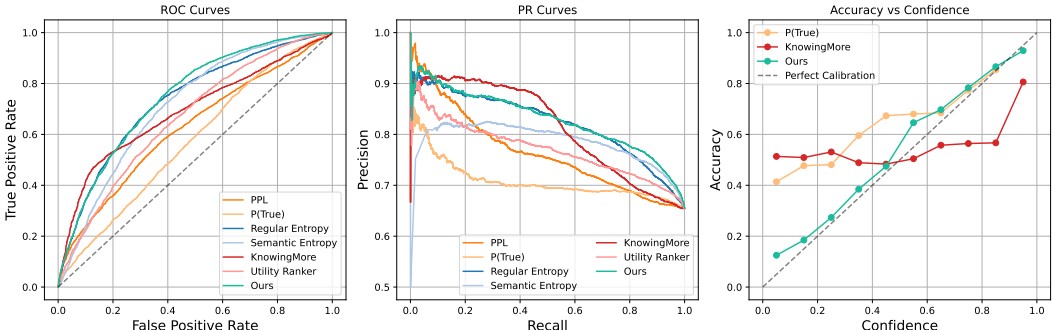

Figure 5: Comparison across various uncertainty estimation methods for language models, illustrated through ROC Curves, PR Curves, and an Accuracy vs. Confidence calibration plot.

We further plot the experiment results in Figure 5. For the ROC curves, a trajectory closer to the top-left corner reflects stronger discriminative ability in distinguishing correct from incorrect responses generated by the language model. In contrast, the PR curves highlight performance on the minority class (incorrect responses), where a curve closer to the top-right corner indicates better precision–recall trade-offs. As illustrated, our method consistently yields larger areas under both curves, confirming its superior performance relative to the baselines. Furthermore, among calibration-based approaches, the confidence scores from the proposed method are well calibrated, closely following the perfect calibration reference line, while P(True) and KnowingMore exhibit noticeable deviations.

### 5.3 Impact of ranker choice on UQ performance

In this work, we rely on Shapley values and the Qwen-3-Reranker-8B model to obtain the relevance layout of context tokens, as introduced in Section 4.4.1. However, this layout should be regarded as an estimated ground truth rather than an absolute one. Since our notion of simulatability is constructed upon this estimate, our method is inherently dependent on the quality of the ranking model. Enhancing the accuracy and robustness of the ground truth estimator through more powerful rerankers, human annotation, or hybrid approaches would directly strengthen the validity of our method. Such reliance on external large language models is not unique to our work but represents a common limitation of existing UQ methods. For example, Semantic Entropy depends critically on the effectiveness of the applied clustering model (e.g., DeBERTa-large (He et al., 2021)) to group outputs with equivalent semantic meaning. We assess our reliance on the external ranker by replacing Qwen-3-Reranker-8B with MSMARCO-MiniLM-L12-v2 (Reimers & Gurevych, 2019) and BGE-v2-m3 (Chen et al., 2024), which are substantially smaller models. As reported in Table 2, our method maintains competitive performance with no substantial degradation. Furthermore, we develop human-annotated datasets to measure the potential bias from reranker models in Appendix G.

Table 2: Comparison of the proposed UQ method's performance when using different ranker models.

| Ranker | AUROC ↑ | AUPRC ↑ | Cali. Acc ↑ | ECE ↓ | Spearman ↑ | Pearson ↑ |
|---|---|---|---|---|---|---|
| MiniLM-L12-v2 (33.4M) | 0.71 | 0.81 | 0.69 | 0.14 | 0.33 | 0.36 |
| BGE-v2-m3 (0.6B) | 0.74 | 0.86 | 0.75 | 0.04 | 0.36 | 0.42 |
| Qwen-3-Reranker-8B | 0.75 | 0.83 | 0.73 | 0.04 | 0.39 | 0.45 |

## 6 Conclusion

In this work, we propose a novel uncertainty quantification (UQ) framework for retrieval-augmented LLMs that leverages internal information flow to assess the importance of context tokens in generated responses. By introducing emergence order and contribution layout, along with the concepts of simulatability and concentration, our method captures both the propagation and focus of contextual information within the model. Experimental results on SQuAD 2.0 with LLaMA-3.2-3B-Instruct demonstrate that our approach provides more reliable uncertainty estimates and outperforms existing baselines, highlighting the value of incorporating the transformer's attention mechanism for robust UQ in RAG settings.

ETHICS STATEMENT

All authors have carefully read and agree to abide by the ICLR Code of Ethics. In preparing this work, we have reflected on possible ethical considerations, including issues of fairness, bias, privacy, and potential societal impacts of our methods. We have made every effort to ensure that the research was conducted responsibly and transparently, with appropriate acknowledgment of limitations and scope. We emphasize that this study does not knowingly incorporate data or methods that would compromise the rights, dignity, or safety of individuals or groups. In addition, we have considered potential risks of misuse and have aimed to present our findings in a manner that minimizes the likelihood of harmful applications.

REPRODUCIBILITY STATEMENT

We have taken deliberate steps to enhance the reproducibility of our work. The main text provides a clear description of the models, evaluation protocols, and experimental setup. Where appropriate, we have included further details in the appendix and supplementary materials to ensure that independent researchers can replicate and verify our findings. Assumptions and methodological choices are stated explicitly, and standard practices are followed to ensure comparability with prior work. Hyperparameters, evaluation criteria, and other implementation details are carefully documented to reduce ambiguity. Together, these measures are intended to support reproducibility, transparency, and scientific rigor, while allowing the community to build upon and validate our contributions.

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

# A    THE USE OF LARGE LANGUAGE MODELS

We acknowledge the use of a large language model (ChatGPT, OpenAI) for editorial assistance. Its role was limited to improving the readability of the manuscript by smoothing phrasing and correcting grammar. The research ideas, methodology, theoretical results, experiments, and technical writing were entirely conducted and authored by the researchers.

# B    ADDITIONAL EXPERIMENT RESULTS

We provide additional experiment results on SQuAD2.0 (Rajpurkar et al., 2018), HotpotQA (Yang et al., 2018), and MS MARCO (Nguyen et al., 2016) using Llama-3.2-3B-Instruct (Grattafiori et al., 2024) and Gemma-3-4B-it Team (2025). Besides, we add two more white-box UQ baselines: Attention Score (Sriramanan et al., 2024) and Focus (Zhang et al., 2023). Performance summaries for both models are reported in Table 3 and Table 4, respectively. Across all datasets and model architectures, the proposed method consistently delivers strong performance, demonstrating its effectiveness for uncertainty quantification (UQ).

Table 3: UQ performance of different methods on three datasets evaluated with the LLaMA-3.2-3B-Instruct model. The estimated relevance layouts are provided by Qwen-3-Reranker-8B. Bold indicates the best value for each metric. The second-best results are underlined.

| Dataset | Method | AUROC | AUPRC | Cali. Acc | ECE | Spearman | Pearson |
|---------|--------|-------|-------|-----------|-----|----------|---------|
| SQuAD2.0 | PPL | 0.622 | 0.770 | / | / | 0.192 | 0.209 |
|  | P(True) | 0.573 | 0.713 | 0.533 | 0.161 | 0.130 | 0.083 |
|  | Regular Entropy | 0.720 | 0.807 | / | / | 0.361 | 0.358 |
|  | Semantic Entropy | 0.714 | 0.784 | / | / | 0.322 | 0.261 |
|  | Attention Score | 0.513 | 0.718 | / | / | 0.032 | 0.022 |
|  | Focus | 0.703 | 0.830 | / | / | 0.336 | 0.364 |
|  | KnowingMore | 0.692 | 0.812 | 0.653 | 0.222 | 0.301 | 0.259 |
|  | Utility Ranker | 0.658 | 0.771 | / | / | 0.283 | 0.217 |
|  | Ours | **0.748** | **0.833** | **0.734** | **0.041** | **0.394** | **0.450** |
| HotpotQA | PPL | 0.582 | 0.912 | / | / | 0.004 | 0.085 |
|  | P(True) | 0.567 | 0.873 | 0.666 | 0.303 | -0.050 | -0.041 |
|  | Regular Entropy | 0.651 | 0.924 | / | / | 0.061 | 0.180 |
|  | Semantic Entropy | 0.614 | 0.911 | / | / | **0.150** | 0.160 |
|  | Attention Score | 0.478 | 0.867 | / | / | -0.044 | -0.031 |
|  | Focus | **0.701** | **0.944** | / | / | 0.125 | 0.196 |
|  | KnowingMore | 0.590 | 0.905 | 0.877 | 0.106 | 0.125 | 0.054 |
|  | Utility Ranker | 0.597 | 0.905 | / | / | 0.059 | 0.115 |
|  | Ours | 0.671 | 0.934 | **0.879** | **0.006** | 0.131 | **0.208** |
| MS MARCO | PPL | 0.592 | 0.691 | / | / | 0.153 | 0.177 |
|  | P(True) | 0.557 | 0.664 | 0.579 | 0.103 | 0.113 | 0.105 |
|  | Regular Entropy | 0.654 | 0.729 | / | / | 0.253 | 0.266 |
|  | Semantic Entropy | 0.528 | 0.611 | / | / | 0.053 | 0.047 |
|  | Attention Score | 0.509 | 0.561 | / | / | -0.070 | -0.072 |
|  | Focus | 0.690 | 0.752 | / | / | 0.303 | 0.305 |
|  | KnowingMore | 0.623 | 0.702 | 0.611 | 0.136 | 0.191 | 0.209 |
|  | Utility Ranker | 0.593 | 0.661 | / | / | 0.122 | 0.167 |
|  | Ours | **0.727** | **0.778** | **0.679** | **0.021** | **0.356** | **0.392** |

Table 4: UQ performance of different methods on three dataset evaluated with the Gemma- 3-4B-it model. The estimated relevance layouts are provided by Qwen-3-Reranker-8B. Bold indicates the best value for each metric. The second-best results are underlined.

| Dataset | Method | AUROC | AUPRC | Cali. Acc | ECE | Spearman | Pearson |
|---------|--------|-------|-------|-----------|-----|----------|---------|
| SQuAD2.0 | PPL | 0.639 | 0.622 | / | / | 0.237 | 0.236 |
| | P(True) | 0.545 | 0.521 | 0.508 | 0.438 | 0.058 | 0.033 |
| | Regular Entropy | 0.658 | 0.633 | / | / | 0.279 | 0.270 |
| | Semantic Entropy | 0.590 | 0.546 | / | / | 0.165 | 0.207 |
| | Attention Score | 0.529 | 0.518 | / | / | 0.044 | 0.047 |
| | Focus | 0.653 | 0.636 | / | / | 0.260 | 0.229 |
| | KnowingMore | 0.620 | 0.625 | 0.590 | 0.259 | 0.200 | 0.213 |
| | Utility Ranker | 0.642 | 0.614 | / | / | 0.213 | 0.252 |
| | Ours | **0.703** | **0.684** | **0.644** | **0.008** | **0.346** | **0.370** |
| HotpotQA | PPL | 0.605 | 0.772 | / | / | 0.140 | 0.181 |
| | P(True) | 0.525 | 0.725 | 0.319 | 0.575 | 0.015 | 0.025 |
| | Regular Entropy | 0.617 | 0.779 | / | / | 0.167 | 0.184 |
| | Semantic Entropy | 0.530 | 0.727 | / | / | 0.059 | 0.097 |
| | Attention Score | 0.507 | 0.698 | / | / | -0.023 | -0.034 |
| | Focus | 0.645 | **0.832** | / | / | 0.192 | 0.197 |
| | KnowingMore | 0.532 | 0.739 | 0.707 | 0.181 | 0.041 | 0.040 |
| | Utility Ranker | 0.545 | 0.744 | / | / | 0.076 | 0.084 |
| | Ours | **0.650** | 0.814 | **0.713** | **0.017** | **0.226** | **0.248** |
| MS MARCO | PPL | 0.561 | 0.495 | / | / | 0.102 | 0.112 |
| | P(True) | 0.548 | 0.532 | 0.568 | 0.359 | 0.093 | 0.086 |
| | Regular Entropy | 0.570 | 0.505 | / | / | 0.127 | 0.124 |
| | Semantic Entropy | 0.574 | 0.528 | / | / | 0.112 | 0.152 |
| | Attention Score | 0.523 | 0.413 | / | / | -0.060 | -0.053 |
| | Focus | 0.574 | 0.519 | / | / | 0.102 | 0.124 |
| | KnowingMore | 0.598 | 0.523 | 0.582 | 0.131 | 0.170 | 0.179 |
| | Utility Ranker | 0.564 | 0.516 | / | / | 0.103 | 0.117 |
| | Ours | **0.706** | **0.627** | **0.651** | **0.062** | **0.342** | **0.369** |

A key advantage of our information-flow-based metrics is their enhanced interpretability, stemming from their reliance on the language model's mechanisms rather than superficial dataset patterns. We posit that this makes our approach inherently more robust to distribution shifts. We validate this claim by applying calibrators trained on one dataset to test data from entirely different distributions (see Table 5 and Table 6). Crucially, we observe that different calibrators perform similarly on a given test set. The consistency across training sources provides strong evidence that our method generalizes well, mitigating the common problem of sensitivity to train-test distribution shifts.

Table 5: Generalizability of the proposed method using Llama-3.2-3B-Instruct. The estimated relevance layouts are generated by Qwen-3-Reranker-8B.

| | SQuAD2.0 (Test) | | HotpotQA (Test) | | MS MARCO (Test) | |
|---|---|---|---|---|---|---|
| | AUROC | AUPRC | AUROC | AUPRC | AUROC | AUPRC |
| SQuAD2.0 (Train) | 0.748 | 0.833 | 0.658 | 0.930 | 0.695 | 0.759 |
| HotpotQA (Train) | 0.728 | 0.838 | 0.671 | 0.934 | 0.692 | 0.752 |
| MS MARCO (Train) | 0.715 | 0.833 | 0.633 | 0.923 | 0.727 | 0.778 |

Table 6: Generalizability of the proposed method using Gemma-3-4B-it. The estimated relevance layouts are generated by Qwen-3-Reranker-8B.

| | SQuAD2.0 (Test) | | HotpotQA (Test) | | MS MARCO (Test) | |
| | AUROC | AUPRC | AUROC | AUPRC | AUROC | AUPRC |
|---|---|---|---|---|---|---|
| SQuAD2.0 (Train) | 0.703 | 0.684 | 0.623 | 0.801 | 0.655 | 0.576 |
| HotpotQA (Train) | 0.663 | 0.652 | 0.650 | 0.814 | 0.672 | 0.574 |
| MS MARCO (Train) | 0.625 | 0.645 | 0.633 | 0.779 | 0.706 | 0.627 |

## C    COMPUTATION OF RANK-BIASED OVERLAP (RBO)

Rank-Biased Overlap (RBO) (Webber et al., 2010) is a measure of similarity between two ranked lists that ranges from 0 to 1 and emphasizes agreement at higher-ranked items. Values closer to 1 indicate that the two lists are highly similar, while values closer to 0 indicate that they are dissimilar.

Consider we have two permutations of the same indices of length $N$: $\pi_X = [x_1, x_2, \ldots, x_N]$ and $\pi_Y = [y_1, y_2, \ldots, y_N]$. At each depth $d = 1, \ldots, N$, compute the overlap between the top-$d$ items:

$$A_d = \{x_1, \ldots, x_d\} \cap \{y_1, \ldots, y_d\}, \quad \text{agr}(d) = \frac{|A_d|}{d}.$$

The RBO score applies a geometric weighting to emphasize higher ranks:

$$\text{RBO}_p(\pi_X, \pi_Y) = (1 - p) \sum_{d=1}^{N} p^{d-1} \text{agr}(d), \quad 0 < p < 1,$$

where $p$ is the persistence parameter controlling the weight decay: larger $p$ assigns more weight to lower-ranked items, while smaller $p$ emphasizes top-ranked positions.

In our work, we apply RBO to compare the ranked index lists $\pi_E, \pi_C, \pi_P$ with the relevance ranking $\pi_R$, capturing the alignment between the model's internal processing and the estimated importance of context tokens. The results of our method in Table 1 are reported with $p = 0.7$.

To evaluate the impact of different RBO hyperparameters, we present the experiment results of $p$ in $\{0.1, 0.3, 0.5, 0.7, 0.9\}$ in Table 7. Varying the RBO hyperparameter does not significantly affect the performance of our method. This robustness arises because correct predictions exhibit strong agreement with the estimated relevance layout across a broad range of context positions, rather than being concentrated only among the top-ranked tokens. Consequently, changing pdoes not substantially alter the relative ordering of examples (correct predictions consistently yield higher RBO scores), so RBO remains a reliable measure of prediction uncertainty across different $p$ values.

Table 7: AUROC and AUPRC of the proposed method with varying RBO hyperparameters. The estimated relevance layouts are generated by Qwen-3-Reranker-8B.

| Model | $p$ | SQuAD2.0 | | HotpotQA | | MS MARCO | |
| | | AUROC | AUPRC | AUROC | AUPRC | AUROC | AUPRC |
|---|---|---|---|---|---|---|---|
| | 0.1 | 0.744 | 0.829 | 0.678 | 0.936 | 0.732 | 0.787 |
| | 0.3 | 0.742 | 0.828 | 0.673 | 0.935 | 0.729 | 0.782 |
| Llama-3.2-3B-Instruct | 0.5 | 0.747 | 0.827 | 0.675 | 0.936 | 0.730 | 0.784 |
| | 0.7 | 0.748 | 0.833 | 0.671 | 0.934 | 0.727 | 0.778 |
| | 0.9 | 0.749 | 0.831 | 0.671 | 0.935 | 0.737 | 0.788 |
| | 0.1 | 0.701 | 0.680 | 0.644 | 0.812 | 0.703 | 0.614 |
| | 0.3 | 0.702 | 0.679 | 0.641 | 0.812 | 0.705 | 0.623 |
| Gemma-3-4B-it | 0.5 | 0.701 | 0.677 | 0.641 | 0.814 | 0.704 | 0.622 |
| | 0.7 | 0.703 | 0.684 | 0.650 | 0.814 | 0.706 | 0.627 |
| | 0.9 | 0.703 | 0.678 | 0.644 | 0.811 | 0.704 | 0.622 |

## D  KL DIVERGENCE FOR DISCRETE PROBABILITY DISTRIBUTIONS

Let $\mu = [\mu_1, \mu_2, \ldots, \mu_N]$ and $\nu = [\nu_1, \nu_2, \ldots, \nu_N]$ denote two discrete probability distributions over $N$ elements, with $\mu_i, \nu_i \geq 0$ and $\sum_{i=1}^{N} \mu_i = \sum_{i=1}^{N} \nu_i = 1$. The Kullback-Leibler (KL) divergence from $\mu$ to $\nu$ is defined as

$$\mathrm{KL}(\mu \,\|\, \nu) = \sum_{i=1}^{N} \mu_i \log \frac{\mu_i}{\nu_i}.$$

In our setting, $\mu$ corresponds to the normalized contribution layouts $\widehat{\mathbf{C}}_{\mathbf{I}}^{\mathrm{layout}}$ and $\widehat{\mathbf{P}}_{\mathbf{I}}^{\mathrm{layout}}$ and $\nu$ corresponds to the uniform distribution over $T_c$ tokens $\mathbf{U} = \left[ \frac{1}{T_c}, \frac{1}{T_c}, \ldots, \frac{1}{T_c} \right]$.

After substitution, we get two KL divergences as

$$\mathrm{KL}(\widehat{\mathbf{C}}_{\mathbf{I}}^{\mathrm{layout}} \,\|\, \mathbf{U}) = \sum_{i=1}^{T_c} \widehat{\mathbf{C}}_i^{\mathrm{layout}} \log \left( \widehat{\mathbf{C}}_i^{\mathrm{layout}} T_c \right),$$

$$\mathrm{KL}(\widehat{\mathbf{P}}_{\mathbf{I}}^{\mathrm{layout}} \,\|\, \mathbf{U}) = \sum_{i=1}^{T_c} \widehat{\mathbf{P}}_i^{\mathrm{layout}} \log \left( \widehat{\mathbf{P}}_i^{\mathrm{layout}} T_c \right),$$

These formulations quantify the concentration of the layouts relative to a uniform distribution: higher KL values indicate that the layout is more concentrated on a small subset of tokens, whereas lower KL values indicate a more uniform distribution of importance across tokens.

## E  MULTI-LEVEL GRANULARITY

To capture a more comprehensive picture of how context information is processed within the model, we analyze emergence order and contribution layouts at multiple levels of granularity: token (subword)-level, word-level, and phrase-level.

Token (subword)-level provides the most fine-grained view, directly reflecting the internal representation of the model's vocabulary. Since many language models operate on subword units (e.g., Byte Pair Encoding or SentencePiece), examining this level allows us to trace how the model assembles meaning from its smallest representational units.

Word-level aggregates contributions and emergence orders across all subwords belonging to the same word. This reduces fragmentation introduced by subword tokenization, making the analysis more interpretable and directly comparable to human linguistic intuitions about words.

Phrase-level further groups words into coherent multi-word expressions. This clustering is conducted based on Shapley values in the relevance layout Lundberg & Lee (2017a), which quantify each token's marginal contribution to the overall interpretation. By aggregating words that consistently share high relevance and interact strongly in terms of contribution, the phrase-level representation captures compositional semantics that cannot be observed at the word level alone. This granularity allows us to study how the model organizes meaning across larger linguistic units.

## F  FEATURE DISCRIMINATIVE ANALYSIS

We evaluate the discriminative ability of each feature—simulatability measures at token-, word-, and phrase-level, concentration measures, and the context relevance score $r$—with respect to the labels. Table 8 reports the AUROC, AUPRC, Spearman correlation, and Pearson correlation of each feature on SQuAD2.0 dataset, evaluated using LLaMA-3.2-3B-Instruct for inference and Qwen-3-Reranker-8B for relevance estimation.

From the results, we observe that the features perform similarly across most metrics. For example, simulatability measures at word- and token-levels achieve comparable AUROC and AUPRC, while

the relevance score $r$ slightly outperforms the others. This relative uniformity in performance suggests that no single feature is sufficiently decisive on its own to estimate the quality of the model's response reliably.

These findings motivate the development of a calibrator that aggregates all features into a unified confidence estimate. By combining simulatability, concentration, and relevance, the calibrator leverages complementary information captured at different granularities and from different aspects of context and model behavior, thereby producing a more robust and informative measure of confidence than any individual feature alone.

Table 8: Discriminative ability of features measured by AUROC, AUPRC, Spearman, and Pearson coefficients, evaluated using LLaMA-3.2-3B-Instruct for inference and Qwen-3- Reranker-8B for relevance estimation. Similar performance motivates a calibrator for robust confidence estimation.

| Metric | | AUROC ↑ | AUPRC ↑ | Spearman ↑ | Pearson ↑ |
|---|---|---|---|---|---|
| token-level | $\text{RBO}(\pi_{\mathbf{E}}, \pi_{\mathbf{R}})$ | 0.56 | 0.70 | 0.13 | 0.10 |
| | $\text{RBO}(\pi_{\mathbf{C}}, \pi_{\mathbf{R}})$ | 0.59 | 0.72 | 0.16 | 0.14 |
| | $\text{RBO}(\pi_{\mathbf{P}}, \pi_{\mathbf{R}})$ | 0.59 | 0.72 | 0.16 | 0.14 |
| word-level | $\text{RBO}(\pi_{\mathbf{E}}, \pi_{\mathbf{R}})$ | 0.60 | 0.73 | 0.17 | 0.14 |
| | $\text{RBO}(\pi_{\mathbf{C}}, \pi_{\mathbf{R}})$ | 0.60 | 0.73 | 0.17 | 0.17 |
| | $\text{RBO}(\pi_{\mathbf{P}}, \pi_{\mathbf{R}})$ | 0.61 | 0.73 | 0.19 | 0.18 |
| phase-level | $\text{RBO}(\pi_{\mathbf{E}}, \pi_{\mathbf{R}})$ | 0.54 | 0.69 | 0.09 | 0.07 |
| | $\text{RBO}(\pi_{\mathbf{C}}, \pi_{\mathbf{R}})$ | 0.60 | 0.72 | 0.18 | 0.15 |
| | $\text{RBO}(\pi_{\mathbf{P}}, \pi_{\mathbf{R}})$ | 0.59 | 0.72 | 0.17 | 0.15 |
| $\text{KL}(\widehat{\mathbf{C}}_{\mathbf{I}}^{\text{layout}} \parallel \mathbf{U})$ | | 0.63 | 0.75 | 0.22 | 0.22 |
| $\text{KL}(\widehat{\mathbf{P}}_{\mathbf{I}}^{\text{layout}} \parallel \mathbf{U})$ | | 0.64 | 0.75 | 0.23 | 0.23 |
| relevance score $r$ | | 0.67 | 0.76 | 0.23 | 0.26 |

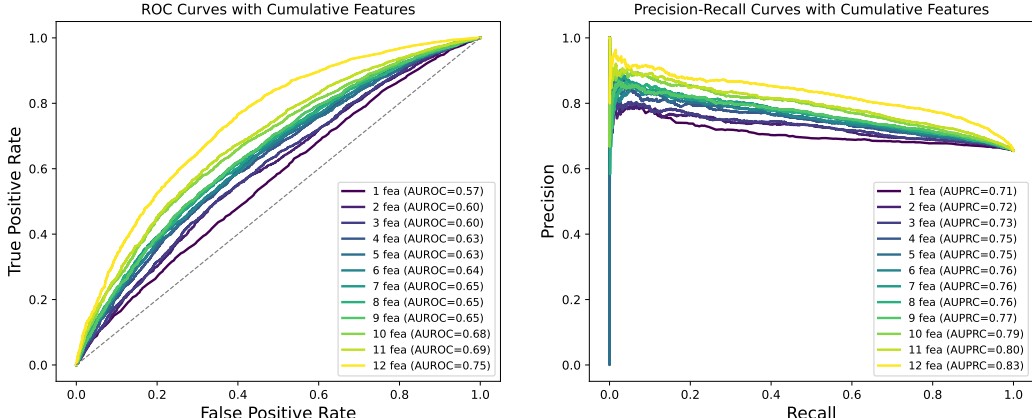

Figure 6: ROC and Precision-Recall curves showing the calibrated confidence performance as features are added cumulatively on SQuAD 2.0 dataset. Each line represents the first $k$ features in the order listed in Table 8, illustrating how discriminative ability progressively improves as more features are incorporated into the calibrator. LLaMA-3.2-3B-Instruct is used for inference and Qwen-3-Reranker-8B is applied for relevance estimation.

Figure 6 illustrates the impact of cumulatively adding features on the calibrated confidence performance. As shown, each line represents the model's performance using the first $k$ features in the order specified in Table 8. Both the ROC and Precision-Recall curves demonstrate a clear trend: as more features are incorporated into the calibrator, the discriminative ability steadily improves. Early features contribute substantially to performance gains, while later features provide incremental improvements, highlighting the complementary information captured by different feature types. We can observe that the relevance score $r$ (the last added feature) leads to a notable improvement, particularly in the high false positive rate (FPR) region of the ROC curve and the high recall region of the PR curve. This can be explained by its role in quantifying how informative the context is

for answering the question. In the high-FPR region of the ROC curve, the model tends to produce many false positives alongside true positives; by incorporating $r$, the calibrator can down-weight predictions that are supported by less relevant context, reducing spurious positive predictions and improving discrimination. Similarly, in the high-recall region of the PR curve, many true positives are already being retrieved, but false positives remain prevalent. Here, $r$ helps the calibrator assign lower confidence to predictions with weak contextual support, effectively lowering false positives while preserving most true positives. In both cases, $r$ allows the confidence estimate to better separate informative from uninformative predictions, leading to pronounced improvement precisely in these challenging regions of the curves.

However, Figure 6 also raises a concern that most of the gains may come from the relevance score $r$, rather than from the proposed information-flow features. To isolate their contributions, we conduct a clean ablation study comparing (i) our information-flow features and (ii) a "relevance-score-only" setup, using Qwen-3-Reranker-8B as the reranker model. The results are summarized in Table 9. We observe that the proposed features consistently outperform the reranker-only setup across all datasets, demonstrating that the observed performance gains primarily stem from the proposed information-flow metrics rather than the reranker itself.

Table 9: AUROC comparison between the proposed information-flow features and reranker-score-only setups across datasets, showing that most performance gains come from the proposed features. Qwen-3-Reranker-8B is applied for relevance estimation. Higher values are shown in bold.

| Model | dataset | information-flow-features-only | relevance-score-only |
|---|---|---|---|
| LLaMA-3.2-3B-Instruct | SQuAD2.0 | **0.693** | 0.585 |
| | HotpotQA | **0.650** | 0.504 |
| | MS MARCO | **0.709** | 0.601 |
| Gemma-3-4B-it | SQuAD2.0 | **0.676** | 0.593 |
| | HotpotQA | **0.639** | 0.506 |
| | MS MARCO | **0.617** | 0.529 |

## G  RERANKER MODEL BIAS MEASUREMENT BY HUMAN-ANNOTATION

We verify the estimated relevance layouts from Qwen-3-Reranker-8B and measure the extent of bias from the reranker model. Specifically, for each dataset, we randomly sample 500 examples. Using the relevance layouts produced by Qwen-3-Reranker-8B as a reference, annotators inspect the top-ranked tokens for each example and label a layout "correct" if those top-ranked tokens were indeed helpful for answering the query. We retain only the examples with correct layouts. Then, we evaluate the simulatability metrics of Llama-3.2-3B-Instruct on both the retained subset and the original 500-sample collection for each dataset. The performance difference illustrate how much bias is introduced from the reranker model.

AUROC and AUPRC of simulatability metrics on original and retained samples are shown in Table 10 and Table 11, which also list the percentage of samples retained after verification. We observe that AUROC and AUPRC increase slightly after human verification in general, indicating that the reranker's estimated relevance layouts contain some bias. This bias arises from the absence of true golden relevance layout, rather than from our information-flow method. Moreover, the improvements are not significantly large, showing that the reranker remains a practical choice, especially when automatically processing large-scale data.

Table 10: Comparison of **AUROC** for simulatability metrics on the original 500 samples versus the human-verified samples. Higher values are in bold.

| | Metric | SQuAD2.0 | | HotpotQA | | MS MARCO | |
|---|---|---|---|---|---|---|---|
| | | Original | Retained (94%) | Original | Retained (90%) | Original | Retained (84%) |
| token-level | $RBO(\pi_E, \pi_R)$ | 0.600 | **0.612** | 0.574 | **0.602** | 0.658 | **0.675** |
| | $RBO(\pi_C, \pi_R)$ | **0.603** | 0.590 | 0.534 | **0.535** | 0.657 | **0.668** |
| | $RBO(\pi_P, \pi_R)$ | **0.634** | 0.631 | 0.566 | **0.602** | 0.616 | **0.620** |
| word-level | $RBO(\pi_E, \pi_R)$ | 0.592 | **0.595** | 0.551 | **0.582** | 0.599 | **0.603** |
| | $RBO(\pi_C, \pi_R)$ | **0.551** | 0.550 | **0.530** | 0.518 | **0.613** | 0.611 |
| | $RBO(\pi_P, \pi_R)$ | 0.652 | **0.655** | 0.540 | **0.585** | 0.593 | **0.600** |
| phrase-level | $RBO(\pi_E, \pi_R)$ | 0.631 | **0.632** | 0.557 | **0.600** | 0.676 | **0.707** |
| | $RBO(\pi_C, \pi_R)$ | 0.621 | **0.630** | 0.575 | **0.580** | 0.706 | **0.726** |
| | $RBO(\pi_P, \pi_R)$ | 0.666 | **0.668** | 0.560 | **0.615** | 0.659 | **0.685** |

Table 11: Comparison of **AUPRC** for simulatability metrics on the original 500 samples versus the human-verified samples. Higher values are in bold.

| | Metric | SQuAD2.0 | | HotpotQA | | MS MARCO | |
|---|---|---|---|---|---|---|---|
| | | Original | Retained (94%) | Original | Retained (90%) | Original | Retained (84%) |
| token-level | $RBO(\pi_E, \pi_R)$ | 0.931 | **0.942** | 0.914 | **0.929** | 0.700 | **0.781** |
| | $RBO(\pi_C, \pi_R)$ | 0.936 | **0.941** | 0.904 | **0.914** | 0.702 | **0.779** |
| | $RBO(\pi_P, \pi_R)$ | 0.939 | **0.945** | 0.903 | **0.929** | 0.675 | **0.744** |
| word-level | $RBO(\pi_E, \pi_R)$ | 0.934 | **0.944** | 0.900 | **0.918** | 0.645 | **0.715** |
| | $RBO(\pi_C, \pi_R)$ | **0.945** | 0.940 | 0.892 | **0.901** | 0.659 | **0.713** |
| | $RBO(\pi_P, \pi_R)$ | 0.949 | **0.955** | 0.893 | **0.923** | 0.643 | **0.718** |
| phrase-level | $RBO(\pi_E, \pi_R)$ | 0.942 | **0.948** | 0.908 | **0.929** | 0.720 | **0.794** |
| | $RBO(\pi_C, \pi_R)$ | 0.945 | **0.956** | 0.905 | **0.917** | 0.756 | **0.817** |
| | $RBO(\pi_P, \pi_R)$ | 0.952 | **0.958** | 0.902 | **0.930** | 0.707 | **0.777** |

# H  BUILT-IN MONOTONICITY OF ONE-DIMENSIONAL UNCERTAINTY REPRESENTATIONS

We recognize that the inclusion of a calibrator in our method could lead to the question of whether performance gains stem from the proposed metrics or the post-processing. To address this directly, we designed our evaluation to dissociate these two factors. We equipped baselines whose original formulations do not involve training with the same calibration procedure used in our method.

We report results for both their raw scores and their calibrated variants in Table 12 and Table 13. The results show that the calibrated variants of these baselines do not outperform their raw versions, and in most cases perform even worse due to overfitting. This outcome is expected: these baselines inherently produce a single scalar uncertainty score (e.g., Perplexity, Semantic Entropy) that is intrinsically designed to correlate with prediction error monotonically. In other words, their discriminative power is largely "built-in". Applying a calibrator to such one-dimensional signals offers no new information and, as our results show, often degrades performance through overfitting.

In contrast, our method and other multi-dimensional approaches (e.g., Utility Ranker, Knowing-More) generate a spectrum of complementary indicators. Conventional evaluation frameworks like AUROC, which require a single scalar, are inherently ill-suited to assess these multi-dimensional signals directly. The post-hoc model is therefore not a performance-enhancing "calibrator" but a necessary scalarization function. Its role is to project the rich, multi-faceted information from our metrics onto a single axis for fair comparison. Thus, this step is not a privileged addition but a fundamental requirement to make multi-dimensional UQ methods evaluable against their scalar counterparts.

Table 12: AUROC and AUPRC results of baselines on SQuAD2.0, HotpotQA, and MS MARCO using Llama-3.2-3B-Instruct. Higher values are in bold.

| | | SQuAD2.0 | | HotpotQA | | MS MARCO | |
|---|---|---|---|---|---|---|---|
| | | AUROC | AUPRC | AUROC | AUPRC | AUROC | AUPRC |
| PPL | Raw | **0.622** | **0.770** | **0.582** | **0.912** | **0.592** | **0.691** |
| | Cali. | 0.619 | 0.763 | 0.580 | 0.910 | 0.590 | 0.684 |
| P(True) | Raw | **0.573** | **0.713** | 0.567 | 0.873 | **0.557** | **0.664** |
| | Cali. | 0.570 | 0.710 | **0.600** | **0.879** | 0.500 | 0.592 |
| Regular Entropy | Raw | **0.720** | **0.807** | **0.651** | **0.924** | **0.654** | **0.729** |
| | Cali. | 0.719 | 0.805 | 0.649 | 0.920 | 0.651 | 0.722 |
| Semantic Entropy | Raw | **0.714** | **0.784** | **0.614** | **0.911** | **0.528** | **0.611** |
| | Cali. | 0.710 | 0.773 | 0.500 | 0.879 | 0.521 | 0.606 |
| Attention Score | Raw | **0.513** | **0.718** | 0.478 | 0.867 | 0.509 | 0.561 |
| | Cali. | 0.506 | 0.712 | **0.512** | **0.882** | **0.539** | **0.618** |
| Focus | Raw | **0.703** | **0.830** | **0.701** | **0.944** | **0.690** | **0.752** |
| | Cali. | 0.700 | 0.828 | 0.699 | 0.940 | 0.663 | 0.739 |

Table 13: AUROC and AUPRC results of baselines on SQuAD2.0, HotpotQA, and MS MARCO using Gemma-3-4B-it. Higher values are in bold.

| | | SQuAD2.0 | | HotpotQA | | MS MARCO | |
|---|---|---|---|---|---|---|---|
| | | AUROC | AUPRC | AUROC | AUPRC | AUROC | AUPRC |
| PPL | Raw | **0.639** | **0.622** | **0.605** | **0.772** | **0.561** | **0.495** |
| | Cali. | 0.638 | 0.617 | 0.604 | 0.768 | 0.551 | 0.487 |
| P(True) | Raw | **0.545** | **0.521** | **0.525** | **0.725** | **0.548** | **0.532** |
| | Cali. | 0.500 | 0.493 | 0.500 | 0.712 | 0.500 | 0.481 |
| Regular Entropy | Raw | **0.658** | **0.633** | 0.617 | **0.779** | 0.570 | 0.505 |
| | Cali. | 0.657 | 0.629 | **0.618** | 0.778 | **0.587** | **0.511** |
| Semantic Entropy | Raw | **0.590** | **0.546** | 0.530 | 0.727 | 0.574 | **0.528** |
| | Cali. | 0.500 | 0.494 | **0.533** | **0.728** | **0.575** | 0.526 |
| Attention Score | Raw | **0.529** | **0.518** | 0.507 | 0.698 | 0.523 | 0.413 |
| | Cali. | 0.512 | 0.502 | **0.522** | **0.733** | **0.526** | **0.453** |
| Focus | Raw | 0.653 | 0.636 | **0.645** | **0.832** | **0.574** | **0.519** |
| | Cali. | **0.654** | **0.643** | 0.644 | 0.827 | 0.573 | 0.516 |

# I EXAMPLES

## I.1 INPUT SEQUENCE EXAMPLE

An illustrative example of the input sequence is shown below, with fixed words marked in bold to clearly delineate them from the varying portions across different samples.

> **Answer the question in no more than five words.**
> **Context:** Two Polish friends in Paris were also to play important roles in Chopin's life there. His fellow student at the Warsaw Conservatory, Julian Fontana, had originally tried unsuccessfully to establish himself in England; Albert Grzymala, who in Paris became a wealthy financier and society figure, often acted as Chopin's adviser and "gradually began to fill the role of elder brother in his life." Fontana was to become, in the words of Michalowski and Samson, Chopin's "general factotum and copyist."
> **Question:** What familial role was Albert Grzymała compared to in regard to Frédéric?
> **Answer:**

### I.2 PREDICTION CORRECTNESS DETERMINATION EXAMPLE

We provide a concrete example below to illustrate how we determine whether a model's response is correct using Qwen2.5-7B (Qwen et al., 2025). Suppose the model generates the statement "The capital of Washington state is Seattle."

> **Convert the following Q&A into a single factual sentence.**
> **Question:** Where is the capital of Washington state?
> **Answer:** Seattle.
> **Statement:**

We also construct a reference statement based on the ground-truth answer "Olympia," namely "The capital of Washington state is Olympia." We then use HHEM-2.1-Open (Bao et al., 2024) to assess whether the predicted statement is incorrect.

Crucially, this evaluation considers the semantics of the question, rather than relying on surface token overlap. Some questions naturally admit multiple correct answers. For example, for the question "When did World War II break out?", both statements "World War II broke out in 1939." and "World War II broke out in late 1930s." are valid. This semantic-level assessment is therefore essential for accurately determining correctness.

### I.3 MANHATTAN-DISTANCE-BASED CONTRIBUTION EXAMPLE

In Eq. (5), we compute the Manhattan distance between $a(\mathbf{y}_i, \mathbf{x}_j)$ and $\mathbf{y}_i$ as

$$\|\mathbf{y}_i - a(\mathbf{y}_i, \mathbf{x}_j)\|_1. \tag{13}$$

Subsequently, we subtract this distance from $\|\mathbf{y}\|_1$

$$\|\mathbf{y}_i\|_1 - \|\mathbf{y}_i - a(\mathbf{y}_i, \mathbf{x}_j)\|_1, \tag{14}$$

and apply a rectification by taking the maximum with zero:

$$\max\left(0, \|\mathbf{y}_i\|_1 - \|\mathbf{y}_i - a(\mathbf{y}_i, \mathbf{x}_j)\|_1\right). \tag{15}$$

This operation yields a positive value only when the Manhattan distance between $a(\mathbf{y}_i, \mathbf{x}_j)$ and $\mathbf{y}_i$ is sufficiently small. Specifically, if the vectors are in close proximity, the distance in Eq. (13) is small, and the resulting difference in Eq. (14) is positive. Conversely, if the vectors are widely separated, the difference becomes negative and is clipped to zero.

Thus, this formulation defines a similarity measure bounded by a Manhattan-distance threshold, where $\|\mathbf{y}_i\|_1$ establishes the maximum permissible distance. The similarity decreases linearly with increasing distance and vanishes entirely when the distance exceeds the specified threshold.

To visualize the behavior of this metric, consider a two-dimensional setting where $\mathbf{y}_i = (1, 1)$. We evaluate Eq. (15) across a grid of vectors $a(\mathbf{y}_i, \mathbf{x}_j)$. Figure 7 produces a diamond-shaped region of positive values centered at the target $(1, 1)$, reflecting the geometry of the L1 norm. Outside this diamond (i.e., when the Manhattan distance exceeds 2), the score is exactly zero, illustrating the effect of the

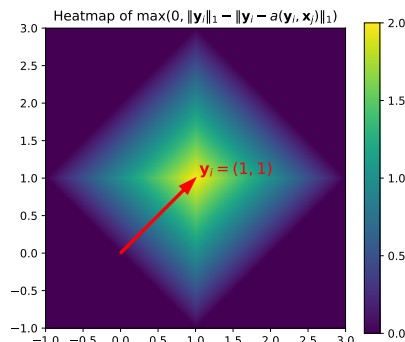

Figure 7: Heatmap of Manhattan-distance-based similarity between $a(\mathbf{y}_i, \mathbf{x}_j)$ and $\mathbf{y}_i = (1, 1)$.

hard cutoff. This example provides an intuitive geometric interpretation of the metric: the level sets indicate the extent to which $a(\mathbf{y}_i, \mathbf{x}_j)$ remains sufficiently close to the target $\mathbf{y}_i$ before the deviation surpasses the allowable threshold.

## J   LIMITATIONS

### J.1   CONSTRAINED APPLICATION SCOPE

The primary limitation of our method stems from its nature as a white-box approach, which requires access to internal model representations. Consequently, it is not directly applicable to closed-source large language models (LLMs) where such access is restricted. This reflects a fundamental trade-off between interpretability and universality in the current LLM ecosystem.

Despite this, the value of our work is threefold. First, it provides a level of mechanistic insight into uncertainty that is unattainable with black-box and gray-box methods, offering a valuable benchmark for understanding the origins of model uncertainty. Second, it is immediately applicable to the growing suite of powerful open-source models, which are critical for academic research, safety auditing, and transparent deployments. Finally, the framework established here lays a foundation for future research into gray-box techniques that might approximate these information-theoretic measures with more limited access. Future work will focus on extending this paradigm. We aim to develop hybrid approaches that can operate effectively in gray-box settings and to explore distillation techniques to transfer the interpretability of white-box uncertainty estimates.

### J.2   COMPUTATION COST

A primary consideration for our method is the memory cost associated with computing the token contribution matrices $C^{(l)} \in \mathbb{R}^{T \times T}$ for each layer $l$. The peak memory consumption occurs during the computation for a single layer. This step requires storing a raw, lower-triangular embedding matrix of size $T \times T \times d$, leading to a memory complexity of $O(T^2 d)$. Crucially, since intermediate embeddings for each layer can be discarded after processing, this peak memory cost is independent of the total number of layers $L$. We identify two practical strategies to mitigate this $O(T^2 d)$ cost:

(1) Low-Rank Approximation: The dimensionality $d$ of each vector in the matrix can be substantially reduced via projection, effectively lowering the $d$ factor in the $O(T^2 d)$ complexity.

(2) Sparse Storage: The token contribution matrices are typically lower-triangular and often exhibit sparsity, as many off-diagonal entries are negligible. Storing only the significant values can dramatically reduce the memory footprint.

After computation, storing the final projected matrices $C^{(l)}$ for all $L$ layers requires only $O(T^2 L)$ memory. Given that $L \ll d$ in standard language model architectures, this cost is negligible compared to the peak computational overhead. This final storage requirement corresponds to the cost of computing the product of these matrices for the overall analysis.

## K   FAITHFULNESS DEMONSTRATION

To move beyond correlation and establish the causal faithfulness of the identified information flows, we conducted a controlled ablation study. Specifically, we randomly selected 100 correctly predicted samples from each experimental configuration. For each sample, we used the three proposed measures, namely $\bar{\mathbf{E}}_{\mathbf{I}}$, $\bar{\mathbf{C}}_{\mathbf{I}}^{\text{layout}}$, and $\bar{\mathbf{P}}_{\mathbf{I}}^{\text{layout}}$, to identify the most and least critical context tokens. We then performed two ablation procedures: one where we ablated the five top-ranked tokens, and another where we ablated the five bottom-ranked tokens, before re-running inference.

The results, detailed in Table 14, Table 15, and Table 16, demonstrate a consistent and sharp contrast. Ablating the top-ranked tokens causes a majority (over 50%) of the previously correct predictions to become incorrect. Conversely, ablating the bottom-ranked tokens results in negligible performance degradation. This stark difference in model sensitivity provides direct causal evidence that the tokens ranked highly by our method are functionally necessary for the model's correct reasoning. This confirms that our information-flow extraction method identifies tokens that genuinely drive predictions, rather than those that merely correlate with correct outcomes.

Table 14: Number of correct predictions after ablation based on $\bar{\mathbf{E}}_{\mathbf{I}}$.

| ablation | SQUAD2 | | MS MARCO | | HotpotQA | |
|---|---|---|---|---|---|---|
| | top | bottom | top | bottom | top | bottom |
| Llama-3.2-3B-Instruct | 48 | 98 | 35 | 96 | 37 | 98 |
| Gemma-3-4B-it | 26 | 98 | 19 | 96 | 29 | 99 |

Table 15: Number of correct predictions after ablation based on $\bar{\mathbf{C}}_{\mathbf{I}}^{\text{layout}}$.

| ablation | SQUAD2 | | MS MARCO | | HotpotQA | |
|---|---|---|---|---|---|---|
| | top | bottom | top | bottom | top | bottom |
| Llama-3.2-3B-Instruct | 32 | 96 | 30 | 97 | 39 | 95 |
| Gemma-3-4B-it | 32 | 96 | 24 | 94 | 25 | 97 |

Table 16: Number of correct predictions after ablation based on $\bar{\mathbf{P}}_{\mathbf{I}}^{\text{layout}}$.

| ablation | SQUAD2 | | MS MARCO | | HotpotQA | |
|---|---|---|---|---|---|---|
| | top | bottom | top | bottom | top | bottom |
| Llama-3.2-3B-Instruct | 42 | 96 | 37 | 95 | 34 | 96 |
| Gemma-3-4B-it | 39 | 98 | 33 | 95 | 22 | 96 |

