# OpenReview forum: "Information Flow Reveals When to Trust Language Models"
_ICLR.cc/2026/Conference — Submitted to ICLR 2026_

### Official Review · Reviewer_Cx2r · 2025-10-25

**Soundness:** 1
**Presentation:** 2
**Contribution:** 3
**Rating:** 4
**Confidence:** 3

**Summary:**

The paper proposes an uncertainty-aware confidence estimator for RAG that looks inside a transformer to quantify how much each context token contributes to a predicted answer. The authors compose per-layer contribution matrices to attribute the answer to specific context tokens, yielding two signals—simulatability (alignment with a reranker’s token-level relevance) and concentration (sharpness of support). A light calibrator combines these with the reranker score to predict confidence. On LLaMA-3.2-3B-Instruct model, it outperforms standard UQ baselines and shows better calibration.

**Strengths:**

* The core idea of estimating confidence by tracing the model’s internal information flow rather than just looking at outputs or inputs is interesting, and the token-level explanations are intuitive to read and discuss.
* The two signals work well together: alignment with an external relevance view plus how narrowly the model concentrates its support, and this combo translates into clear gains in discrimination and calibration on a standard QA benchmark.

**Weaknesses:**

* The core signal (“simulatability”) is defined by how closely the model’s internal attributions match a reranker’s token relevance, so both the metric and the final predictor depend on the reranker—creating a dependency loop that inherits the reranker’s biases. Some potential tests:
    - In addition to §5.3, swap in a very different reranker or two and see how much the confidence quality moves.
    - Build a small human-labeled “relevance layout” set and compare the model’s simulatability to humans vs. to the reranker.
    - Vary the rank-agreement hyperparameters (e.g., how top-heavy the agreement metric is) and see if conclusions flip.
* A large share of the reported gains may come from the reranker’s single scalar score rather than the new information-flow features; the paper does not show a clean ablation of “reranker-only” vs. “proposed features without the reranker,” which is needed to prove incremental value.
* The evaluation is narrowly scoped.
    - The number and variety of models tested are very limited.
    - The authors only use very short answers on single-passage SQuAD) and never tests a true end-to-end RAG setup with multi-document retrieval and distractors, so it’s unclear whether the benefits hold in realistic settings.
     - Baselines are not on equal footing: the proposed method uses a tuned post-hoc calibrator while several baselines appear “as is.” Applying the same calibrator to all methods (and reporting both raw and calibrated results) would make the comparison fair.
* Faithfulness is assumed rather than demonstrated; there are no causal tests (e.g., ablating top-ranked vs. low-ranked tokens) to show that the extracted token flows actually drive correctness rather than merely correlate with it.

**Questions:**

* What are the runtime/memory costs for building/composing contribution matrices at 2k–8k context lengths? How does the method affect inference speed/latency?

---

> ### Author Response · Authors · 2025-11-19
>
> We thank the reviewers for their valuable feedback. We have conducted extensive additional experiments and revisions in direct response to the comments, detailed in our **newly uploaded manuscript**. Our point-by-point responses below reference the new results and updated sections to demonstrate how each concern has been addressed.
>
> >The number and variety of models tested are very limited.
>
> In response to the request for a comprehensive evaluation, we have conducted extensive experiments on three benchmark datasets, **SQuAD2.0**, **HotpotQA**, and **MS MARCO**, using two model architectures, **Llama-3.2-3B-Instruct** and **Gemma-3-4B-it**.
>
> The results, presented in **Tables 3** and **Table 4** in **Appendix B**, demonstrate that our proposed method consistently delivers strong performance. It achieves the best or second-best result across all dataset-model combinations. This consistent, top-tier performance underscores the effectiveness of our information-flow-based approach for uncertainty quantification.
>
> >The core signal (“simulatability”) is defined by how closely the model’s internal attributions match a reranker’s token relevance, so both the metric and the final predictor depend on the reranker—creating a dependency loop that inherits the reranker’s biases. In addition to §5.3, swap in a very different reranker or two and see how much the confidence quality moves.
>
> We additionally conduct experiments using another reranker model, **BGE-v2-m3 (0.6B)**. Its results, along with those of the two rerankers included in the main paper, are shown in **Table 2**. We observe that the performance of the proposed method remains stable, exhibiting no significant fluctuation when applied with different reranker models.
>
> >Build a small human-labeled “relevance layout” set and compare the model’s simulatability to humans vs. to the reranker.
>
> To quantitatively measure this bias, we constructed human-verified "golden relevance layout" collections for evaluation. Specifically, for each dataset, we randomly sampled 500 examples. Using the relevance layouts produced by the Qwen-3-Reranker-8B as a provisional reference, human annotators inspected the top-ranked tokens for each example. A layout was labeled "correct" only if these tokens were objectively helpful for answering the query. We then computed our simulatability metrics on two sets: the original 500-sample collection and the verified subset of correct layouts.
>
> The results, detailed in **Table 10** and **Table 11** in **Appendix G**, show that AUROC and AUPRC values increase slightly on the human-verified subset in general. This confirms that the reranker's estimated layouts do contain a degree of bias, which we attribute to the fundamental challenge of the absence of a true golden standard for relevance, rather than to our information-flow method itself.
>
> Crucially, the observed improvements are modest, demonstrating that while imperfect, the reranker provides a practical approximation for large-scale automated analysis.
>
> > Vary the rank-agreement hyperparameters (e.g., how top-heavy the agreement metric is) and see if conclusions flip.
>
> The sensitivity of our method to the Rank-Biased Overlap (RBO) hyperparameter $p$ was evaluated across a range of values from 0.1 to 0.9. As shown in **Table 7** in **Appendix C**, which is based on relevance layouts from the Qwen-3-Reranker-8B, the performance remains stable despite these variations.
>
> This robustness arises because correct predictions exhibit strong agreement with the estimated relevance layout across a broad range of context positions, rather than being concentrated only among the top-ranked tokens. Consequently, changing $p$ does not substantially alter the relative ordering of examples (correct predictions consistently yield higher RBO scores), so RBO remains a reliable measure of prediction uncertainty across different $p$ values.
>
> >A large share of the reported gains may come from the reranker’s single scalar score rather than the new information-flow features; the paper does not show a clean ablation of “reranker-only” vs. “proposed features without the reranker,” which is needed to prove incremental value.
>
> To isolate the gain of the proposed information-flow features from that of the reranker’s scalar relevance score, we conduct a clean ablation comparing “proposed features without the reranker” and “reranker-only” setups, using Qwen-3-Reranker-8B as the reranker model. The results are summarized in **Table 9** in **Appendix F**. We observe that the proposed features consistently outperform the reranker-only setup across all datasets, demonstrating that the observed performance gains primarily stem from the proposed information-flow metrics rather than the reranker itself.

---

> ### Author Response · Authors · 2025-11-19
>
> >The authors only use very short answers on single-passage SQuAD) and never tests a true end-to-end RAG setup with multi-document retrieval and distractors, so it’s unclear whether the benefits hold in realistic settings.
>
> We thank the reviewer for this valuable comment. Our current experiments focus on short-form QA, which serves as a standard and foundational setting for evaluating UQ methods; many prior works adopt the same formulation [1,2,3]. Extending uncertainty quantification to long-form generation is indeed more challenging, particularly for information-flow–based analysis, because later generated tokens depend heavily on earlier ones, which may themselves be unreliable. This introduces compounding uncertainty that requires new methodological advances. We fully agree that this is an important next step and consider it a promising direction for future work.
>
> Regarding the concern about real-world applicability, we hope the new experimental results on **HotpotQA** and **MS MARCO** help address this point. Both datasets include multiple passages and distractors, which naturally simulate the noisy outputs of real retrieval systems. The strong performance of our method on these settings demonstrates its effectiveness under conditions that more closely resemble practical RAG pipelines.
>
> [1] Perez-Beltrachini, Laura, and Mirella Lapata. "Uncertainty Quantification in Retrieval Augmented Question Answering." arXiv preprint arXiv:2502.18108 (2025).
>
> [2] Shujian Zhang, Chengyue Gong, and Eunsol Choi. 2021. Knowing More About Questions Can Help: Improving Calibration in Question Answering. In Findings of the Association for Computational Linguistics: ACL-IJCNLP 2021, pages 1958–1970, Online. Association for Computational Linguistics.
>
> [3] Si, Chenglei, et al. "Re-examining calibration: The case of question answering." arXiv preprint arXiv:2205.12507 (2022).
>
> >What are the runtime/memory costs for building/composing contribution matrices at 2k–8k context lengths?
>
> We thank the reviewer for raising this important point about memory complexity.  Assume an LLM with hidden dimension $d$, sequence length $T$, and $L$ layers. To obtain $\textbf{C}^{(l)}∈\mathbb{R}^{T×T}$ for a single layer $l$, we first store a raw $T×T×d$ lower-triangular embedding matrix (lower-triangular along the $T×T$ dimensions). The memory required is $O(T^2⋅d)$. We do not need to store all layers simultaneously. After computing the contribution matrix for a layer, intermediate embeddings can be discarded. Therefore, the peak memory is $O(T^2⋅d)$, independent of $L$.  If d=2048, for 2 k tokens, this requires 8.2 GB of memory, which is manageable. For 8 k tokens, the requirement grows to about 130 GB. Practical strategies can mitigate the cost. For example, low-rank approximation can reduce the dimensionality of each vector stored in the matrix, lowering the $d$ factor in $O(T^2⋅d)$. Also, sparse storage can exploit the fact that many entries in the lower-triangular matrix are extremely small or negligible, storing only significant values.
>
> Storing the $\textbf{C}^{(l)}∈\mathbb{R}^{T×T}$ for all $L$ layers requires $O(L⋅T^2)$ memory, which is the cost for computing their product. As $L≪d$, this memory cost is much lower than $O(T^2⋅d)$.
>
> We included the discussion on memory complexity in **Appendix J.2** of the revised manuscript.
>
> >How does the method affect inference speed/latency?
>
> For a sequence of 2k tokens with 2048-dimensional embeddings and 28 layers, computing all lower-triangular Manhattan distances requires ~8.2 × 10⁹ FLOPs, which takes only a few milliseconds on a 10 TFLOPS GPU. In contrast, multiplying the 28 dense 2000×2000 contribution matrices requires ~2.2 × 10¹¹ FLOPs, adding roughly 50–100 ms of latency.
> For 8k tokens, Manhattan distance computation scales to ~1.3 × 10¹¹ FLOPs, taking ~50–100 ms in practice, while the product of 28 dense 8000×8000 matrices requires ~1.4 × 10¹³ FLOPs, dominating runtime with an estimated 2s.
>
> Therefore, the overall latency is dominated by the final composition step, not the initial Manhattan distance computation. We consider this a worthwhile trade-off, as it provides a complete, interpretable uncertainty estimate that is computationally feasible for a wide range of practical sequence lengths.

---

> ### Author Response · Authors · 2025-11-19
>
> >Baselines are not on equal footing: the proposed method uses a tuned post-hoc calibrator while several baselines appear “as is.” Applying the same calibrator to all methods (and reporting both raw and calibrated results) would make the comparison fair.
>
> To address the reviewer’s concern, we performed an additional experiment for all baselines whose original formulation does not involve training. We created a “calibrated” version by applying the same post-hoc learning procedure used in our method. We report results for both their raw scores and their calibrated variants in **Table 12** and **Table 13** in **Appendix H**.
>
> The results show that the calibrated variants of these baselines do not outperform their raw versions, and in most cases perform even worse due to overfitting. This outcome is expected: these baselines inherently produce a single scalar uncertainty score that is already constructed to be monotonically correlated with prediction reliability. In other words, **their discriminative power is largely “built-in”**.  Applying an additional calibrator cannot introduce new information and is likely to distort the existing signal.
>
> In contrast, our method and multi-dimensional baselines, such as Utility Ranker and KnowingMore, generate a vector of complementary uncertainty indicators. These multi-dimensional signals cannot be meaningfully evaluated under AUROC or AUPRC, which require a scalar uncertainty score. A post-hoc model is therefore not an optional enhancement but a necessary scalarization mechanism that combines the different dimensions into a single usable metric. This step does not give our method an advantage over baselines; rather, it is a way to evaluate a multi-dimensional UQ method within conventional scoring frameworks.
>
> >Faithfulness is assumed rather than demonstrated; there are no causal tests (e.g., ablating top-ranked vs. low-ranked tokens) to show that the extracted token flows actually drive correctness rather than merely correlate with it.
>
> To assess the causal faithfulness of our extracted token flows, we conduct an ablation analysis. Specifically, we randomly select 100 correctly predicted samples from each experimental setup and use each of the three layouts, namely $\mathbf{\bar{E}}_ {\mathbf{I}}$, $\mathbf{\bar{C}}_ {\mathbf{I}}^ {\text{layout}}$, $\mathbf{\bar{P}}_ {\mathbf{I}}^ {\text{layout}}$, to identify the top- and bottom-ranked context tokens. We then ablate the five top-ranked and five bottom-ranked tokens, respectively, and re-run inference.
>
> As shown in **Table 14**, **Table 15**, and **Table 16** in **Appendix K**, ablating the top-ranked tokens causes more than half of the previously correct predictions to become incorrect, whereas ablating bottom-ranked tokens yields negligible performance degradation. This result provides causal evidence that the identified top-ranked tokens indeed drive correct predictions rather than merely correlate with them.

---

> ### Comment · Reviewer_Cx2r · 2025-11-24
>
> Thanks the authors for the very substantial set of new experiments and clarifications. The added evaluations across more datasets and additional models, reranker variants, hyperparameter studies, and ablation analyses improve the empirical grounding of the paper.
>
> Given these additions and the strengthened empirical evidence, I have raised my score.

---

> > ### Author Response · Authors · 2025-11-25
> >
> > Thank you for your thoughtful review and for appreciating the additional experiments and clarifications we provided. We are glad that the expanded empirical evidence addressed your concerns and are grateful for the improved score.

---

### Official Review · Reviewer_o6ip · 2025-10-30

**Soundness:** 2
**Presentation:** 3
**Contribution:** 3
**Rating:** 4
**Confidence:** 3

**Summary:**

In this submission, the authors introduce a series of novel criteria that offer a fresh perspective on uncertainty quantification for LLMs. By leveraging established concepts such as information flow and contribution matrices, the authors present a framework for assessing the relative importance of input tokens with respect to model outputs. The proposed measurement captures both the emergence order and contribution layout of input tokens, thereby providing deeper insight into the relationship between inputs and outputs and enabling an estimation of response confidence.
The experimental results include comparisons between the proposed method and several existing UQ baselines on a specific dataset. The findings suggest the proposed method exhibits certain advantages over baselines.

**Strengths:**

- The authors introduce a novel perspective by leveraging information flow to reveal the importance of input tokens.
- The workflow for calculating contribution matrices and deriving the subsequent uncertainty quantification criteria is clearly described and easy to follow.
- The experimental results demonstrate the improvements of the proposed method compared to baselines.

**Weaknesses:**

- The motivation for applying information flow in uncertainty quantification is not clear. The authors are encouraged to provide further discussion on this point. For example, what specific advantages are brought by introducing information flow, and what makes the proposed method outperform existing studies in uncertainty quantification tasks? Does applying information flow improve the generalizability and robustness of uncertainty quantification, or make it applicable to a wider range of scenarios?
- While the proposed method is demonstrated in the context of RAG, the experimental setup is limited to a single base model and one dataset. As a result, the experiments in this paper are somewhat limited and not fully convincing.
- The writing of this submission should be further polished. For example, the inconsistent usage of $\mathbf{y_i}$ and $\mathbf{y}_i$ in Equation 5.

**Questions:**

Please refer to the weaknesses.

---

> ### Author Response · Authors · 2025-11-19
>
> We thank the reviewer for their insightful feedback. We have conducted extensive additional experiments and revisions in direct response to the comments, detailed in our **newly uploaded manuscript**. Our point-by-point responses below reference the new results and updated sections to demonstrate how each concern has been addressed.
>
> >While the proposed method is demonstrated in the context of RAG, the experimental setup is limited to a single base model and one dataset. As a result, the experiments in this paper are somewhat limited and not fully convincing.
>
> In response to the request for a comprehensive evaluation, we have conducted extensive experiments on three benchmark datasets, **SQuAD2.0**, **HotpotQA**, and **MS MARCO**, using two model architectures, **Llama-3.2-3B-Instruct** and **Gemma-3-4B-it**.
>
> The results, presented in **Tables 3** and **Table 4** in **Appendix B**, demonstrate that our proposed method consistently delivers strong performance. It achieves the best or second-best result across all dataset-model combinations. This consistent, top-tier performance underscores the effectiveness of our information-flow-based approach for uncertainty quantification.
>
> >The motivation for applying information flow in uncertainty quantification is not clear. The authors are encouraged to provide further discussion on this point. For example, what specific advantages are brought by introducing information flow, and what makes the proposed method outperform existing studies in uncertainty quantification tasks?
>
> Traditional uncertainty quantification (UQ) metrics, such as semantic entropy and perplexity, provide only coarse, global confidence estimates derived directly from a model's output signals (e.g., logits). Consequently, they are frequently ineffective in scenarios where large language models (LLMs) generate incorrect answers with high confidence, which is a core challenge documented in hallucination research [1]. Our method addresses this fundamental limitation by moving beyond surface-level signals. By tracing the internal information flow, we directly inspect which specific parts of the input the model attends to and quantify the influence of each individual token on the final prediction. This fine-grained, token-level contribution signal enables the detection of flawed or unreliable reasoning pathways even when the model's overall confidence appears deceptively high.
>
> [1] Ziwei Ji, Nayeon Lee, Rita Frieske, Tiezheng Yu, Dan Su, Yan Xu, Etsuko Ishii, Ye Jin Bang, Andrea Madotto, and Pascale Fung. 2023. Survey of Hallucination in Natural Language Generation. ACM Comput. Surv. 55, 12, Article 248 (December 2023), 38 pages. https://doi.org/10.1145/3571730
>
> >Does applying information flow improve the generalizability and robustness of uncertainty quantification, or make it applicable to a wider range of scenarios?
>
> Because our method is rooted in the model’s internal computation process, it naturally provides stronger interpretability and **greater robustness to out-of-distribution (OOD) shifts**. To further assess this robustness, we report cross-dataset evaluation results in **Table 5** and **Table 6** in **Appendix B**. We observe that calibrators trained on different sources yield highly similar performance when evaluated on the same test distribution, indicating that our interpretability-driven UQ method is not sensitive to the distribution shift between training and test data.
>
> Furthermore, our experiments on HotpotQA evaluate the method on multi-hop reasoning tasks, as the dataset includes both bridge-type and comparison-type questions requiring to combine information from multiple pieces of evidence. In addition, SQuAD2.0 and MS MARCO contain distractor passages, which naturally simulate noisy retriever outputs, thereby proving the effectiveness of our method under **a wider range of scenarios**.
>
> > The writing of this submission should be further polished. For example, the inconsistent usage of  y_i and y_i in Equation 5.
>
> Thank you for pointing this out. We modified the typos in the updated document.

---

> > ### Comment · Reviewer_o6ip · 2025-11-26
> >
> > Thank you for conducting additional experiments to demonstrate the effectiveness and robustness of the proposed methods. These results have addressed some of my concerns.
> >
> > Regarding the motivation for applying information flow, the provided explanation does not offer much beyond what was already included in the original submission. It remains unclear to me what specific advantages this approach brings, as information flow appears to be just one among various parallel methods.

---

> ### Author Response · Authors · 2025-11-26
> **Motivation and Novelty of Information-Flow–Based Uncertainty**
>
> We appreciate the reviewer’s concern that the motivation for using information flow may appear similar to other mechanism-based uncertainty approaches. However, our method is substantially different from prior *‘parallel’* white-box baselines in both scope and capability, and we clarify this below.
>
> **(1) Existing white-box methods use only partial or aggregated internal signals, whereas our approach traces the full causal computation path.**
>
> The prior work [1] uses attention matrices and hidden states on **a single chosen layer**. However, a strong attention signal to a context token can be diminished through a subsequent value-matrix multiplication, and a hidden state with high log-generalized variance at one layer can be substantially attenuated by downstream layers. *Thus, relying on a single layer overlooks how signals evolve across the full forward pass.*
>
> Similarly, [2] uses **max-pooling across all layers** to aggregate attention matrices. However, this aggregation collapses the entire layerwise computation into a single statistic, discarding the ordering and transformation through the network. Max-pooling cannot distinguish whether a large attention value arises early or late in the model, or whether its effect is amplified or reduced by later computations. *As a result, pooling obscures the causal chain of how tokens influence the final prediction, losing key information necessary for understanding uncertainty propagation.*
>
> In contrast, our method reconstructs the **complete forward information flow** from input tokens to the final output, sequentially accounting for (i) token-to-token attention interactions, (ii) value-matrix retrieval, and (iii) attention-residual mixing across all layers. This yields **a full, sequential, layer-by-layer trace**, where the context is transformed and propagated throughout the model, and the accumulated result is used to quantify each context token’s contribution to the final prediction.
>
> Beyond this technical novelty, tracking the full path also enables **interpretability**: it identifies which irrelevant tokens lead to incorrect outputs and the precise paths they take, providing diagnostic insights that existing white-box methods cannot achieve.
>
> **(2) Prior mechanism-based methods design metrics only around structural properties, not around contextual usefulness.**
>
> For example, [1] introduces eigenvalue statistics of attention matrices, and [2] uses attention values to weight prediction probabilities. However, these metrics depend **solely on the model’s internal architecture**, without assessing whether the model actually leverages helpful and relevant context when generating an answer.
>
> In contrast, our information-flow formulation explicitly quantifies whether the model’s prediction relies on supportive or irrelevant content in the context. Thus, our uncertainty measure is tied not merely to model mechanics but to **semantic relevance**, which is a dimension that previous work does not evaluate.
>
> **(3) Empirically, our approach yields better uncertainty estimates.**
>
> The two discussed white-box baselines, **Attention Score** [1] and **Focus** [2], are added to **Table 3** and **Table 4**.
> Our experimental results demonstrate that modeling uncertainty via full-path information flow leads to more accurate detection of incorrect responses compared to the existing mechanism-based approaches, confirming that our added architectural and contextual modeling brings real practical benefit.
>
> We hope this explanation clearly demonstrates the specific advantages of our approach over other *‘parallel’* white-box methods, and we welcome any further questions or discussion.
>
> [1] Sriramanan, Gaurang, et al. "LLM-check: Investigating detection of hallucinations in large language models." Advances in Neural Information Processing Systems 37 (2024): 34188-34216.
>
> [2] Tianhang Zhang, Lin Qiu, Qipeng Guo, Cheng Deng, Yue Zhang, Zheng Zhang, Chenghu Zhou, Xinbing Wang, and Luoyi Fu. 2023. Enhancing Uncertainty-Based Hallucination Detection with Stronger Focus. In Proceedings of the 2023 Conference on Empirical Methods in Natural Language Processing, pages 915–932, Singapore. Association for Computational Linguistics.

---

> > ### Author Response · Authors · 2025-11-27
> > **Additional Clarifications — We Look Forward to Your Feedback**
> >
> > Dear Reviewer o6ip,
> >
> > Thank you for your careful reading and insightful comments. In our initial submission and first response, we primarily compared our method to black-box and gray-box approaches, and we appreciate the opportunity to more explicitly **contrast our work with prior white-box (mechanism-based) methods**.
> >
> > We explained why existing white-box methods, which rely on **single-layer snapshots or simple aggregations**, cannot capture the full progression of information through the model. In contrast, our **sequential information-flow formulation** precisely monitors how token embeddings are transformed and propagated throughout the model.
> >
> > We also explicitly assess whether predictions are **grounded in supportive content**, whereas prior mechanism-based works rely solely on **structural statistics of the model**.
> >
> > Finally, our **empirical evaluations** in **Table 3** and **Table 4** against white-box baselines show that our method provides more accurate and reliable uncertainty estimates.
> >
> > We hope this message, together with our previous response, clarifies the unique contributions of our approach and adequately addresses your concerns. We look forward to any further feedback you may have.
> >
> > Best regards,
> >
> > The Authors

---

> > > ### Comment · Reviewer_o6ip · 2025-11-28
> > >
> > > Thank you for your response. I am leaning towards a borderline score (unfortunately, there is no option for a rating of 5). Therefore, I will continue to follow the entire author-review discussion phase and adjust my rating accordingly at the conclusion.

---

### Official Review · Reviewer_63yQ · 2025-10-30

**Soundness:** 3
**Presentation:** 3
**Contribution:** 3
**Rating:** 6
**Confidence:** 2

**Summary:**

The paper proposes an uncertainty quantification framework for RAG that traces the information flow through attention and residual connections. Two main constructs are introduced, emergence order which is a greedy backward extraction of a principal flow that ranks input tokens by the order they enter dominant computation, and contribution layout which is a path composed summary that aggregates token to token contributions across layer.
These are compared against a a token level relevance layout to yield simulatability, and concentration. A learned calibrator combines these with a context-level relevance score to predict answer reliability. They show performance across different benchmarks and show outperforming several output space and input aware UQ baselines.

**Strengths:**

1) The authors present the paper which is methodologically clear and grounded, the paper recasts multi-head attention to define per-layer contribution matrices via an attribution vector and a normalized distance, forming lower-triangular structures consistent with causal masking. This provides a principled basis to trace token interactions across layers
2) A clear, constructive definition of Emergence Order via a greedy backward search over a selection pool yields a ranked, rather than thresholded, importance ordering which improves resolution over prior binary criteria
3) The approach improves AUROC and shows better calibration relative to Perplexity, P(True), (Semantic) Entropy, and two input-aware baselines

**Weaknesses:**

1) All core results are on SQuAD2.0 with a single base model (LLaMA-3.2-3B-Instruct) and short answers. This setting is comparatively “clean,” with relatively local evidence and it lacks representing the multi hop reasoning, long contexts, noisy retrievers and OOD drift, which are all common in RAG applications.
2) The simulatability requires a token-level ground truth relevance layout created by shapley decomposing a rerankers single relevance score. This can create a potential circularity and bias, if the rerankers inductive bias differ from the QA models, simulatability may reward mimicry of the reranker rather than grounding.

**Questions:**

1) Can you also report the computation per-layer contribution matrices and their product, which could be expensive in memory.
2) Results reported are point estimates without confidence intervals, variance, or even seeds. Given a modest AUROC detas, formal uncertainty estimation is essential.
3) Minor typo - Line 224: "embeddning"

---

> ### Author Response · Authors · 2025-11-19
>
> We are grateful for the reviewer's thorough and constructive assessment. In direct response to all points raised, we have performed significant revisions and added new experiments, which are comprehensively detailed in the **newly uploaded manuscript**. Our responses below will clarify our revisions by referencing the updated content, including new tables and analyses.
>
> >All core results are on SQuAD2.0 with a single base model (LLaMA-3.2-3B-Instruct) and short answers.
>
> In response to the request for a comprehensive evaluation, we have conducted extensive experiments on three benchmark datasets, **SQuAD2.0**, **HotpotQA**, and **MS MARCO**, using two model architectures, **Llama-3.2-3B-Instruct** and **Gemma-3-4B-it**.
>
> The results, presented in **Tables 3** and **Table 4** in **Appendix B**, demonstrate that our proposed method consistently delivers strong performance. It achieves the best or second-best result across all dataset-model combinations. This consistent, top-tier performance underscores the effectiveness of our information-flow-based approach for uncertainty quantification.
>
> > This setting is comparatively “clean,” with relatively local evidence and it lacks representing the multi hop reasoning, long contexts, noisy retrievers and OOD drift, which are all common in RAG applications.
>
> We thank the reviewer for this valuable comment. Our new experimental design intentionally incorporates diverse challenges to rigorously evaluate our method. Specifically:
>
> (1). The HotpotQA dataset assesses performance on complex multi-hop reasoning tasks, including both bridge-type and comparison-type questions that require synthesizing information across multiple evidence passages.
>
> (2). Both SQuAD 2.0 and MS MARCO contain distractor passages, which naturally simulate the noisy outputs of real-world retrieval systems. Our method's strong performance in these settings demonstrates its effectiveness in filtering irrelevant information under realistic, challenging conditions.
>
> Regarding robustness to distribution shift, **Table 5** and **Table 6** in **Appendix B** show that calibrators **trained on different sources** achieve consistent performance when evaluated on the **same test distribution**. This indicates that our interpretability-driven UQ method exhibits notable stability and is not highly sensitive to the train-test distribution shift.
>
> Uncertainty quantification in long-text generation is indeed more challenging, particularly when analyzing information flow: later generated tokens are strongly dependent on earlier ones, which themselves may be unreliable. This interdependence complicates faithful uncertainty estimation. We view this as an important and promising direction for future research.
>
> >The simulatability requires a token-level ground truth relevance layout created by shapley decomposing a rerankers single relevance score. This can create a potential circularity and bias, if the rerankers inductive bias differ from the QA models, simulatability may reward mimicry of the reranker rather than grounding.
>
> We acknowledge the reviewer's valid concern regarding potential bias introduced by the reranker. To quantitatively measure this bias, we constructed human-verified "golden relevance layout" collections for evaluation.
>
> For each dataset, we randomly sampled 500 examples. Using the relevance layouts produced by the Qwen-3-Reranker-8B as a provisional reference, human annotators inspected the top-ranked tokens for each example. A layout was labeled "correct" only if these tokens were objectively helpful for answering the query. We then computed our simulatability metrics on two sets: the original 500-sample collection and the verified subset of correct layouts.
>
> The results, detailed in **Table 10** and **Table 11** in **Appendix G**, show that AUROC and AUPRC values increase slightly on the human-verified subsets in general. This confirms that the reranker's estimated layouts do contain a degree of bias, which we attribute to the fundamental challenge of the absence of a true golden standard for relevance, rather than to our information-flow method itself.
>
> Crucially, the observed improvements are modest, demonstrating that while imperfect, the reranker provides a practical approximation for large-scale automated analysis.

---

> ### Author Response · Authors · 2025-11-19
>
> > Can you also report the computation per-layer contribution matrices and their product, which could be expensive in memory.
>
> We thank the reviewer for raising this important point about memory complexity.  Assume an LLM with hidden dimension $d$, sequence length $T$, and $L$ layers. To obtain $\textbf{C}^{(l)}∈\mathbb{R}^{T×T}$ for a single layer $l$, we first store a raw $T×T×d$ lower-triangular embedding matrix (lower-triangular along the $T×T$ dimensions). The memory required is $O(T^2⋅d)$. We do not need to store all layers simultaneously. After computing the contribution matrix for a layer, intermediate embeddings can be discarded. Therefore, the peak memory is $O(T^2⋅d)$, independent of $L$. Practical strategies can mitigate the cost. For example, low-rank approximation can reduce the dimensionality of each vector stored in the matrix, lowering the $d$ factor in $O(T^2⋅d)$. Also, sparse storage can exploit the fact that many entries in the lower-triangular matrix are extremely small or negligible, storing only significant values.
>
> Storing the $\textbf{C}^{(l)}∈\mathbb{R}^{T×T}$ for all $L$ layers requires $O(L⋅T^2)$ memory, which is needed for computing their product. As $L≪d$, this memory cost is much lower than $O(T^2⋅d)$.
>
> We included the discussion on memory complexity in **Appendix J.2** of the revised manuscript.
>
> >Results reported are point estimates without confidence intervals, variance, or even seeds. Given a modest AUROC detas, formal uncertainty estimation is essential.
>
> To ensure stable and reproducible experimental results, we fixed the data sample split using a random seed (42) and performed all inference using greedy decoding. This design eliminates potential variance from data partitioning and generation strategies, isolating the inherent stochasticity of the methods themselves.
>
> Among the evaluated baselines, only Semantic Entropy and the Utility Ranker exhibit stochastic behavior. Semantic Entropy uses an entailment model to cluster semantically similar generations, and this clustering process introduces randomness. Utility Ranker retrieves distractor passages and learns to distinguish useful context through contrastive learning, where the retrieval step is inherently stochastic. Our method also involves randomness because the Shapley-value procedure, which is used to compute relevance layout across context tokens, is a sampling-based estimation.
>
> To quantify this variability, we conducted an analysis on the MS MARCO dataset using the Llama-3.2-3B-Instruct model. We ran both our method and the stochastic baselines three times each. The mean and variance of the resulting AUROC and AUPRC scores are reported in **Table R1** below. From the results, we observe that all variances are extremely small (on the order of 1e-4 to 1e-8), indicating that the experimental outcomes are highly stable. We will perform a full variability analysis across all datasets and models and present these results as soon as the computation is finished.
>
> #### **Table R1. UQ performance on MS MARCO using Llama-3.2-3B-Instruct, reporting the stability across three runs.**
> |          |     AUROC    |     AUPRC    |
> |:---:|:---:|:---:|
> |     Semantic Entropy    |     0.528 (8.04e-8)    |     0.611 (7.19e-8)    |
> |     Utility Ranker    |     0.593 (1.01e-4)    |     0.661 (6.24e-5)    |
> |     Ours    |     0.727 (1.01e-6)    |     0.778 (1.75e-7)    |
>
> >Minor typo - Line 224: "embeddning"
>
> Thank you for pointing this out. We corrected the typo in the latest document.

---

> ### Author Response · Authors · 2025-11-27
> **Additional Evaluation of Experimental Stability**
>
> We have evaluated the variance of our method, Semantic Entropy, and Utility Ranker across three independent runs for all model-dataset combinations, as shown in **Table R2**, **Table R3**, and **Table R4**, respectively. They indicate that the experiment results are stable with negligible variance.
>
> **Table R2. UQ performance of our approach, reporting the stability across three runs (variance in brackets).**
> |     Model | Dataset |     AUROC    |     AUPRC    |
> |:---:|:---:|:---:|:---:|
> |     LLaMA-3.2-3B-Instruct   | SQuAD2.0 |     0.748(7.23e-7)    |     0.833(3.65e-7)    |
> |     LLaMA-3.2-3B-Instruct   | HotpotQA |     0.671(8.61e-7)    |     0.934(5.75e-8)    |
> |     LLaMA-3.2-3B-Instruct   | MS MARCO |     0.727(1.01e-6)    |     0.778(1.75e-7)    |
> |     Gemma-3-4B-it | SQuAD2.0 |     0.703(3.87e-7)    |     0.684(8.81e-8)    |
> |     Gemma-3-4B-it | HotpotQA |     0.650(3.42e-7)    |     0.814(2.75e-8)    |
> |     Gemma-3-4B-it | MS MARCO |     0.706(5.22e-7)    |     0.627(2.90e-6)    |
>
>
> **Table R3. UQ performance of Semantic Entropy, reporting the stability across three runs (variance in brackets).**
> |     Model | Dataset |     AUROC    |     AUPRC    |
> |:---:|:---:|:---:|:---:|
> |     LLaMA-3.2-3B-Instruct   | SQuAD2.0 |     0.714(2.87e-6)    |     0.784(2.99e-6)    |
> |     LLaMA-3.2-3B-Instruct   | HotpotQA |     0.614(5.83e-8)    |     0.911(1.98e-7)    |
> |     LLaMA-3.2-3B-Instruct   | MS MARCO |     0.528(8.04e-8)    |     0.611(7.19e-8)    |
> |     Gemma-3-4B-it | SQuAD2.0 |     0.590(2.10e-5)    |     0.546(2.16e-5)    |
> |     Gemma-3-4B-it | HotpotQA |     0.530(8.67e-4)    |     0.727(4.69e-6)    |
> |     Gemma-3-4B-it | MS MARCO |     0.574(1.41e-6)    |     0.528(2.01e-4)    |
>
> **Table R4. UQ performance of Utility Ranker, reporting the stability across three runs (variance in brackets).**
> |     Model | Dataset |     AUROC    |     AUPRC    |
> |:---:|:---:|:---:|:---:|
> |     LLaMA-3.2-3B-Instruct   | SQuAD2.0 | 0.658(1.44e-6) | 0.771(8.05e-6) |
> |     LLaMA-3.2-3B-Instruct   | HotpotQA | 0.597(1.01e-5) | 0.905(4.54e-7) |
> |     LLaMA-3.2-3B-Instruct   | MS MARCO | 0.593(1.01e-4) | 0.661(6.25e-5) |
> |     Gemma-3-4B-it | SQuAD2.0 | 0.642(1.34e-4) | 0.614(1.55e-4) |
> |     Gemma-3-4B-it | HotpotQA | 0.545(9.08e-5) | 0.744(2.27e-5) |
> |     Gemma-3-4B-it | MS MARCO | 0.564(3.50e-5) | 0.516(4.80e-5) |

---

### Official Review · Reviewer_nenT · 2025-10-31

**Soundness:** 2
**Presentation:** 2
**Contribution:** 2
**Rating:** 2
**Confidence:** 2

**Summary:**

This paper uses mechanistic interpretability sorts of ideas to propose what I believe is a method for white-box confidence estimation, as it relies on having access to the model weights. The work proposes what it terms “simulatability”, which assesses the alignment between context token contributions and their relevance, and “concentration”, which is measure of how many tokens the model relies on for its generation. From my understanding, these can be seen as features for calibrating confidence. The work is positioned as UQ for RAG problems, but I felt that it could apply more broadly. Maybe I did not fully appreciate the RAG aspect in the paper, which is possible. Limited experiments are performed on Squad 2.0 to show how the confidences perform on standard prediction metrics such as AUROC and AUPRC as well as others.

**Strengths:**

While I am not an expert on mechanistic interpretability, I found it interesting to see the process by which features were derived from the underlying attention mechanism of transformers.

The issue of UQ for RAG seems useful but I could not follow the specific challenges in this direction. Perhaps a revised version of the paper can clearly describe the scope of problems that the work is particularly suitable for.

**Weaknesses:**

I found experimentation to be lacking. I understand that it is tricky to do this for many models, particularly large ones, but one model and one dataset is not enough to gauge the value of the method, in my view. I also generally did not fully appreciate the exact way that ground truth was gauged, forming the basis for evaluating confidence of responses. How is an answer deemed correct?

I’m concerned about the choice of baselines and find them somewhat lacking. Did the authors consider using features from logits (like G-NLL from this work: https://arxiv.org/abs/2412.15176) or similarity features (like from this work: https://www.arxiv.org/abs/2510.13836) to calibrate? In other words, what about using other scores/features to train a model with training data? Also, did they consider other white-box methods that use model weights?

I found the method to be quite involved, without enough justification for what was going on with all the transformations, normalizations, distributional distance computations, etc. It was hard to follow. So much is going on and it’s unclear that it’s that valuable in that end – at least based on the evidence in the submission.

Having access to model weights makes the method quite limiting in my view. Much more needs to be mentioned about the severe limitations of the work. There is a reason why there is so much attention in the literature on methods that are either black-box or gray-box (using token logits).

Please note that my limited understanding is reflected in my low confidence. I’m open to changing my opinion as I read other reviews and participate in the discussion.

**Questions:**

Here are some additional comments and questions:

As noted previously, I had a hard time following evaluation. A brief and clear explanation of how correctness of responses is deemed would be appreciated.

I recommend mentioning the limitations of the work in sufficient detail, including the abstract. Issues are raised about competing methods but not about the proposed method, which I think is misleading.

The citation of Vaswani et al. for LLMs is strange. I suggest modifying this line to make the citation about transformers and not LLMs.

The authors say that all the cited work is black-box, but that is not true – some are gray-box methods using logits. Not enough white-box methods, including classic ones on UQ, are cited in the paper.

I had a hard time following the import of the P^{total} computation on page 5, and also much of the content on page 6.

Is the relevance score “r” used for evaluation? On page 7, it seems it is also used as a feature?

---

> ### Author Response · Authors · 2025-11-19
>
> Thank you for your thoughtful and constructive comments. In direct response to your feedback, we have conducted additional experiments and revisions, which are detailed in our **newly uploaded manuscript**. Our responses below will refer to this updated document and its new results to clarify how we have addressed each point.
>
> >I found experimentation to be lacking. I understand that it is tricky to do this for many models, particularly large ones, but one model and one dataset is not enough to gauge the value of the method, in my view.  ......Also, did they consider other white-box methods that use model weights?
> Not enough white-box methods, including classic ones on UQ, are cited in the paper.
>
> In response to the request for a comprehensive evaluation, we have conducted extensive experiments on three benchmark datasets, **SQuAD2.0**, **HotpotQA**, and **MS MARCO**, using two model architectures, **Llama-3.2-3B-Instruct** and **Gemma-3-4B-it**. To ensure a thorough comparison, we also included two additional white-box UQ baselines, **Attention Score** [1] and **Focus** [2].
>
> The results, presented in **Tables 3** and **Table 4** in **Appendix B**, demonstrate that our proposed method consistently delivers strong performance. It achieves the best or second-best result across all dataset-model combinations. This consistent, top-tier performance underscores the effectiveness of our information-flow-based approach for uncertainty quantification.
>
> We would like to explain further why our method outperforms [1] and [2].
>
> The prior work [1] uses attention matrices and hidden states on a single chosen layer. However, a strong attention signal to a context token can be diminished through a subsequent value-matrix multiplication, and a hidden state with high log-generalized variance at one layer can be substantially attenuated by downstream layers. Thus, relying on a single layer overlooks how signals evolve across the full forward pass. Similarly, [2] attempts to account for multiple layers by aggregating attention matrices via max-pooling. However, this method collapses the rich, sequential layer-wise computation into a single statistic. It discards the critical ordering and progressive transformation of information through the network, thereby obscuring the causal chain of how context tokens influence the final prediction.
>
> In contrast, our method yields **a full, sequential, layer-by-layer trace**, where the context is transformed throughout the model, and the accumulated result quantifies each context token’s contribution to the prediction. Moreover, we explicitly assess **whether predictions are grounded in supportive content**, whereas prior mechanism-based works rely solely on structural statistics of the model. The comprehensive architectural and contextual analysis provides key information necessary for understanding uncertainty propagation.
>
> [1] Sriramanan, Gaurang, et al. "LLM-check: Investigating detection of hallucinations in large language models." Advances in Neural Information Processing Systems 37 (2024).
>
> [2] Tianhang Zhang, Lin Qiu, Qipeng Guo, Cheng Deng, Yue Zhang, Zheng Zhang, Chenghu Zhou, Xinbing Wang, and Luoyi Fu. 2023. Enhancing Uncertainty-Based Hallucination Detection with Stronger Focus. In Proceedings of the 2023 Conference on Empirical Methods in Natural Language Processing
>
> >I also generally did not fully appreciate the exact way that ground truth was gauged, forming the basis for evaluating the confidence of responses. How is an answer deemed correct? As noted previously, I had a hard time following the evaluation. A brief and clear explanation of how the correctness of responses is deemed would be appreciated.
>
> We would like to provide a clear description of how correctness is evaluated. Using BERT-based similarity alone may fail to capture the full contextual meaning of a QA pair, so we adopt a two-step semantic evaluation procedure.
>
> (1). **Convert QA pairs into declarative statements.**
> For each example, we prompt Qwen2.5-7B to convert the question and the model’s predicted answer into a single declarative fact (e.g., “Where is the capital of Washington state?” + “Seattle” → “The capital of Washington state is Seattle.”).
> We apply the same conversion to the question and the gold answer.
>
> (2). **Measure semantic alignment.**
> We then feed both declarative statements into HHEM-2.1-Open, which produces a semantic consistency score in [0,1]. A prediction is deemed correct if this score exceeds 0.5.
>
> This approach evaluates answer quality considering the question rather than relying on surface token overlap, and follows common practice in recent LLM evaluation pipelines, such as Section 3.5 of [3]. A more detailed illustrative example is presented in **Appendix I.2**
>
> [3] Qiu, Xin, and Risto Miikkulainen. "Semantic density: Uncertainty quantification for large language models through confidence measurement in semantic space." Advances in neural information processing systems.

---

> ### Author Response · Authors · 2025-11-19
>
> > I’m concerned about the choice of baselines and find them somewhat lacking. Did the authors consider using features from logits (like G-NLL from this work: https://arxiv.org/abs/2412.15176) or similarity features (like from this work: https://www.arxiv.org/abs/2510.13836) to calibrate? In other words, what about using other scores/features to train a model with training data? Issues are raised about competing methods, but not about the proposed method, which I think is misleading.
>
> We thank the reviewer for pointing out the possibility of calibrating on features such as G-NLL [4] or similarity-based scores [5]. However, neither of the referenced works has released their code, which makes it impossible to reproduce their methods or conduct a fair evaluation at this stage.
>
> To address the reviewer’s concern, we performed an additional experiment for all baselines whose original formulation does not involve calibration training. We created a “calibrated” version by applying the same post-hoc learning procedure used in our method. We report results for both their raw scores and their calibrated variants in **Table 12** and **Table 13** in **Appendix H**.
>
> The results show that the calibrated variants of these baselines do not outperform their raw versions, and in most cases perform even worse due to overfitting. This outcome is expected: these baselines inherently produce a single scalar uncertainty score that is already constructed to be monotonically correlated with prediction reliability. In other words, **their discriminative power is largely “built-in”**. Applying an additional calibrator cannot introduce new information and is likely to distort the existing signal.
>
> In contrast, our method and multi-dimensional baselines, such as Utility Ranker and KnowingMore, generate a vector of complementary uncertainty indicators. These multi-dimensional signals cannot be meaningfully evaluated under AUROC or AUPRC, which require a scalar uncertainty score. A post-hoc model is therefore not an optional enhancement but a necessary scalarization mechanism that combines the different dimensions into a single usable metric. This step does not give our method an advantage over baselines; rather, it is a way to evaluate multi-dimensional UQ methods within conventional scoring frameworks.
>
> [4] Aichberger, Lukas, Kajetan Schweighofer, and Sepp Hochreiter. "Rethinking uncertainty estimation in natural language generation." arXiv preprint arXiv:2412.15176 (2024).
>
> [5] Bhattacharjya, Debarun, et al. "SIMBA UQ: Similarity-Based Aggregation for Uncertainty Quantification in Large Language Models." arXiv preprint arXiv:2510.13836 (2025).
>
> >The citation of Vaswani et al. for LLMs is strange. I suggest modifying this line to make the citation about transformers and not LLMs.
>
> We agree that relating the citation directly to the Transformer architecture is more precise. We have modified the text accordingly.
>
> >The authors say that all the cited work is black-box, but that is not true – some are gray-box methods using logits.
>
> You are right to point out that some of the cited works are more accurately described as gray-box methods, as they utilize token-level logits. We have revised descriptions in the manuscript to more precisely characterize the related work.
>
> >Is the relevance score “r” used for evaluation? On page 7, it seems it is also used as a feature?
>
> The relevance score $r$ from the reranker model is used as a feature. The correctness of a prediction is determined by the evaluation model HHEM-2.1-Open, rather than the output $r$ from the reranker model.

---

> ### Author Response · Authors · 2025-11-19
>
> > I found the method to be quite involved, without enough justification for what was going on with all the transformations, normalizations, distributional distance computations, etc. It was hard to follow. So much is going on and it’s unclear that it’s that valuable in that end – at least based on the evidence in the submission. .......I had a hard time following the import of the P^{total} computation on page 5, and also much of the content on page 6.
>
> The core idea of our method is to **quantify how much each context token contributes to the LLM’s response**.
>
> To illustrate the behavior of the Manhattan-distance-based similarity in Eq.(5), we provide a detailed explanation and a toy example in **Appendix I.3** and **Figure 7**.
>
> In Section 4.1, we propose Algorithm 1 to extract the principal information flow matrices $\\{\textbf{P}^{(l)}\\}_{l=1}^L$ and emergence order $\mathbf{E}$ from the last input token's final-layer embeddings. This process refers to **Figure 1 (b)** and **Figure 3**.
>
> In Section 4.2, using the complete flow $\\{\textbf{C}^{(l)}\\}_{l=1}^L$, we compute token-wise contribution by aggregating each token's admissible paths to the final prediction, and obtain $\textbf{C}^\text{layout}$. This process is shown in **Figure 1 (c)** and **Figure 4**.
>
> To provide a more focused view of the significant pathways, we define $\textbf{P}^\text{total}$ on page 5 as the product from the principal flow matrices $\\{\textbf{P}^{(l)}\\}_{l=1}^L$. The construction of $\textbf{P}^\text{total}$ in Eq.(10) directly follows the logic established in Eq.(6), with the key distinction being the substitution of $\textbf{C}^{(l)}$ with $\textbf{P}^{(l)}$. Consequently, the calculation of $\textbf{P}^\text{layout}$ in Eq.(10) is defined analogously to the process described in Eq.(9).
>
> In Section 4.3, as our analysis focuses exclusively on the context tokens, we apply a filtering operation in Eq.(11) to remove contributions unrelated to the context. Besides, a model's response typically comprises multiple generated tokens. Eq.(12) averages across the entire multi-token response. This provides a holistic view of how each piece of context is utilized throughout the generation process.
>
> In Section 4.4.1, we estimate the true usefulness of context tokens. We refer to this estimation as the relevance layout.
>
> In Section 4.4.2, the Rank Biased Overlap (RBO) is used to quantitatively compare the alignment between the estimated relevance layout and our results. Moreover, we posit that a contribution layout that is highly concentrated on a few key tokens indicates a more confident model prediction. To test this, we quantify the concentration of a layout by comparing it to a uniform distribution using Kullback-Leibler (KL) divergence. Since the raw contribution layouts are not constructed as probability distributions, we first apply normalization in Lines 327-328 to ensure a valid and meaningful comparison.
>
> We hope this explanation has successfully conveyed the core logic and rationale behind our approach. We welcome any further questions you may have and are happy to provide additional details.
>
> > Having access to model weights makes the method quite limiting in my view. Much more needs to be mentioned about the severe limitations of the work. There is a reason why there is so much attention in the literature on methods that are either black-box or gray-box (using token logits). I recommend mentioning the limitations of the work in sufficient detail, including the abstract.
>
> We fully acknowledge that our white-box approach, which requires access to internal model representations, is its primary limitation and cannot be directly applied to closed-source LLMs. This indeed represents a fundamental trade-off between depth of interpretability and universal applicability in the current landscape.
>
> However, we believe the value of our work remains significant for three key reasons:
>
> (1). It provides a level of mechanistic insight into uncertainty origins that is simply unattainable through black-box or gray-box methods, establishing a valuable benchmark for the research community.
>
> (2). It is immediately practical and highly relevant for the rapidly growing ecosystem of powerful open-source models, which are crucial for academic research, safety auditing, and applications requiring transparency.
>
> (3). The framework we establish serves as a foundational step toward future gray-box techniques that could approximate our interpretability measures with more limited access, such as logits.
>
> Directly addressing this limitation, our future work will explicitly focus on extending this paradigm. We plan to develop hybrid approaches for gray-box settings and explore distillation techniques to transfer the benefits of white-box uncertainty estimates to broader model classes. We have amended the **abstract** accordingly and expanded upon this important point in a new **Appendix J.1**.

---

### Author Response · Authors · 2025-11-24
**We Welcome and Look Forward to Your Comments**

Dear Reviewers,

We sincerely thank you for your valuable time and insightful comments. Your suggestions were greatly appreciated.

We have substantially revised our manuscript and incorporated a comprehensive set of new experiments, including a detailed ablation study and additional benchmarks as recommended. We believe these additions provide strong empirical support and significantly enhance the robustness of our contributions.

**The updated manuscript is now available for your review.** We hope that these revisions adequately address your concerns, and we are grateful for your constructive feedback, which has helped us further improve the quality of this work.

Best regards,

The Authors

---

### Meta-Review · Area_Chair_e7yJ · 2026-01-06

**Summary:**

The major concerns are partially addressed. Remaining concerns include the paper representation issue and the technical novelty of the proposed method compared to existing white-box methods. These could be addressable but currently the reviewers lean toward borderline.

**Reviewer Concerns:**

**Reviewer nenT**:
The reviewer raised the following five major concerns and they are partially addressed as follows:
1. (*limited experiments*) One model and one dataset is not enough to gauge the value of the method – addressed by evaluating two additional models over three additional datasets, while maintaining the same positive trends.
2. (*missing baselines*) I’m concerned about the choice of baselines and find them somewhat lacking – addressed by adding new results on two white-box methods
3. (*unclear method description*) I found the method to be quite involved … and it’s unclear that it’s that valuable in that end – partially addressed by listing summaries on each step.
4. (*unclear experiment description*) I also generally did not fully appreciate the exact way that ground truth was gauged – provided a short summary.
5. (*limitation on the white-box setup*) Having access to model weights makes the method quite limiting in my view – acknowledge the limitation with a reasonable justification that the white-box methods are still needed for open source models.


The major outstanding concern is the paper representation and the paper’s novelty compared to existing whitebox models. The method section is still hard to follow as it just lists components of the proposed method and does not provide how to use this method at the end (even after the revision). The comparison to existing whitebox methods is missing in the main paper and does not clearly contrast the proposed method to the existing whitebox methods, which undermines the novelty of the proposed method. These issues are partially addressed in the responses but not sufficient.

**Reviewer 63yQ**:
The reviewer raised four major concerns and they are addressed as follows:
1. (*limited experiments and baselines*) All core results are on SQuAD2.0 with a single base model (LLaMA-3.2-3B-Instruct) and short answers, which is relatively weak compared over a not practical setup – addressed by evaluating two additional models over three additional datasets and adding evaluation over a HotpotQA dataset for multi-hop reasoning, while maintaining the same positive trends.
2. (*concern on the simulatability metric*) simulatability may reward mimicry of the reranker rather than grounding – acknowledge the limitation by providing empirical evidence, while claiming that this bias is modest.
3. (*computational overhead*) Can you also report the computation per-layer contribution matrices and their product, which could be expensive in memory – provided the valid analysis on memory complexity in Appendix J.2.
4. (*no random experiments*) Results reported are point estimates without confidence intervals, variance, or even seeds – provided the variation over three random experiments.

No outstanding remaining concerns.



**Reviewer o6ip**:
The reviewer has two major concerns and they are partially addressed as follows:
1. (*unclear motivation on information flow*) The motivation for applying information flow in uncertainty quantification is not clear – explained but it is unsatisfactory as it does not provide motivation in introducing information flow with respect to other white box methods.

2. (*limited evaluation datasets and models*) the experimental setup is limited to a single base model and one dataset – addressed by evaluating two additional models over three additional datasets, while maintaining the same positive trends.

One outstanding concern: the motivation in introducing information flow compared to other white-box methods is missing. The paper is mainly framed without considering existing white-box methods, which falsely mislead readers.


**Reviewer Cx2r**:

1. (*bias on simulatability*) The core signal (“simulatability”) is defined by how closely the model’s internal attributions match a reranker’s token relevance, so both the metric and the final predictor depend on the reranker – addressed by providing additional experiments (proposed by the reviewer) and showing empirical evidence that mitigates the bias from rerankers.

2. (*concern on the effect of information flow*) A large share of the reported gains may come from the reranker’s single scalar score rather than the new information-flow features – provided an ablation study showing the effect of information flow.

3. (*limited evaluation datasets, models, and setups*) The evaluation is narrowly scoped – addressed by evaluating two additional models over three additional datasets and one multi-hop QA dataset, while maintaining the same positive trends.

4. (*inference speed*) How does the method affect inference speed/latency? – provided empirical evidence that the proposed method is practical enough.

No outstanding remaining concerns.

**Reviewer Scores:**

**Reviewer nenT**:
Final expected rating: 4 / final expected confidence: 2 – The concerns are partially addressed and the final manuscript is not enough to mitigate the raised concerns, but the reviewer might increase the rating from 2 to 4 as the remaining concerns could be further addressable.


**Reviewer 63yQ**:
Final expected rating: 6 / final expected confidence: 2 – The concerns are addressed and given the confidence, I expect that the reviewer would not confidently increase its rating.

**Reviewer o6ip**:
Final expected rating: 4 / final expected confidence: 3 – The main concern on the motivation on introducing information flow is still missing, so I expect the reviewer would maintain scores.

**Reviewer Cx2r**:
Final expected rating: 6 / final expected confidence: 3 – The main concerns are all addressed and actually the reviewer confirms to raise the rating from 4 to 6, which looks reasonable to me as well.

---

### Decision · Program_Chairs · 2026-01-26

Reject